# SegGen: Supercharging Segmentation Models with Text2Mask and Mask2Img Synthesis

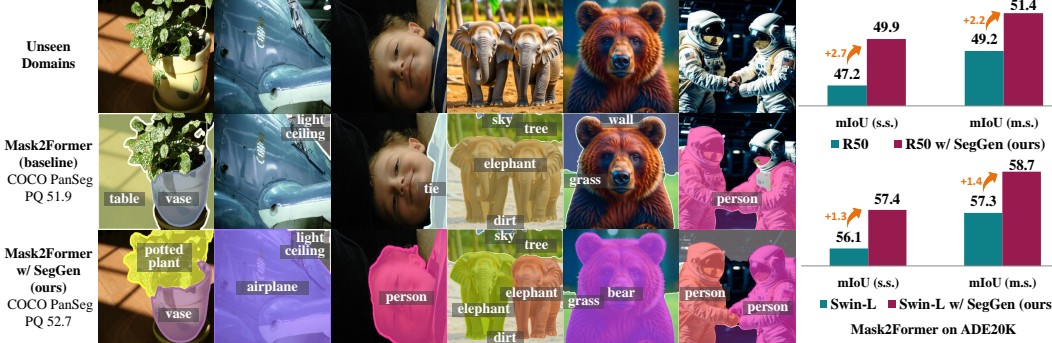

Figure 1: **Effectiveness of SegGen:** Through training with synthetic data generated by the proposed SegGen, we significantly boost the performance of state-of-the-art segmentation model Mask2Former (Cheng et al., 2022) on evaluation benchmarks including ADE20K (Zhou et al., 2016) and COCO (Lin et al., 2014), whilst making it more robust towards challenging images from other domains (the three columns on the left are from PASCAL (Everingham et al., 2015); the three on the right are synthesized by the text-to-image generation model Kandinsky 2 (Forever, 2023)).

## Abstract

We propose SegGen, a highly-effective training data generation method for image segmentation, which pushes the performance limits of state-of-the-art segmentation models to a significant extent. SegGen designs and integrates two data generation strategies: MaskSyn and ImgSyn. (i) MaskSyn synthesizes new mask-image pairs via our proposed text-to-mask generation model and mask-to-image generation model, greatly improving the diversity in segmentation masks for model supervision; (ii) ImgSyn synthesizes new images based on existing masks using the mask-to-image generation model, strongly improving image diversity for model inputs. On the highly competitive ADE20K and COCO benchmarks, our data generation method markedly improves the performance of state-of-the-art segmentation models in semantic segmentation, panoptic segmentation, and instance segmentation. Notably, in terms of the ADE20K mIoU, Mask2Former R50 is largely boosted from 47.2 to 49.9 (**+2.7**); Mask2Former Swin-L is also significantly increased from 56.1 to 57.4 (**+1.3**). These promising results strongly suggest the effectiveness of our SegGen even when abundant human-annotated training data is utilized. Moreover, training with our synthetic data makes the segmentation models more robust towards unseen domains. The project will be open-source upon paper acceptance to promote further study.

## 1 Introduction

Image segmentation explores the identification of objects in visual inputs at the pixel level. Based on the different emphases on category and instance membership information, researchers have divided image segmentation into several tasks (Long et al., 2015; Chen et al., 2015; Kirillov et al., 2019; Qi et al., 2022). For example, semantic segmentation studies pixel-level understanding of object categories, instance segmentation focuses on instance grouping of pixels, while panoptic segmentation considers both. For all these segmentation tasks, obtaining high-quality annotation is challenging as every individual pixel requires human labeling, and a single image can contain millions of pixels. Therefore, compared to other public datasets like ImageNet-21K (with around 14M images),

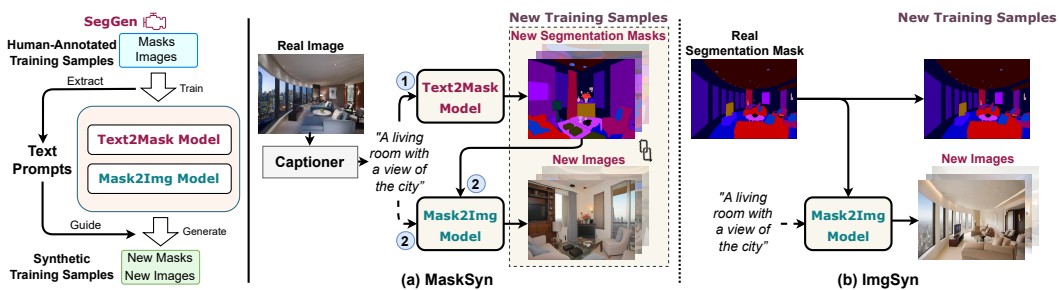

Figure 2: **Illustration of the workflow of our proposed SegGen.** We introduce two generative models: a text-to-mask (Text2Mask) generation model and a mask-to-image (Mask2Img) generation model, based on which we design two approaches for synthesizing segmentation training samples: MaskSyn and ImgSyn. **(a)** MaskSyn focuses on generating new segmentation masks. It first extracts the caption of the real image as a text prompt and uses it to generate new masks with the Text2Mask model. Then, the new masks and text prompt are fed into the Mask2Img model to produce the corresponding new images. **(b)** ImgSyn focuses on the synthesis of new images. It directly inputs human-labeled masks and text prompts into the Mask2Img model to generate new images.

the prevailing human-annotated segmentation datasets are notably smaller. For example, ADE20K dataset (Zhou et al., 2017) contains about 20K images in its training split, while COCO (Lin et al., 2014) has around 118K training images. Although there has been significant development in the structure of segmentation models (Zhao et al., 2017; Li et al., 2019; Zhang et al., 2021a; Cheng et al., 2021; Ye & Xu, 2023), the limited size of training data hinders further performance enhancements and results in inadequate generalization ability to handle images from unfamiliar domains, such as those from other scenes or synthesized by generative models as shown in Fig. 1.

Inspired by the recent success of image generation (Karras et al., 2019; Dhariwal & Nichol, 2021), researchers start to explore using generative models to enhance image segmentation (Baranchuk et al., 2022). A representative direction of this research focuses on synthesizing segmentation training data in a cost-effective manner (Li et al., 2022). These methods rely on the intermediate features of image generation models for segmentation mask generation. Specifically, their synthetic segmentation masks are obtained by either learning segmentation modules on the intermediate features, as in DatasetGAN (Zhang et al., 2021b) and Grounded Diffusion (Li et al., 2023e), or utilizing post-processing methods on the intermediate features, as in DiffuMask (Wu et al., 2023b). The quality of masks generated from the image generation-specific intermediate features is unsatisfactory because the generation models are not trained for image segmentation. Therefore, while they achieve encouraging success in settings with artificially-limited training data, their methods fail to deliver notable enhancements on standard benchmarks in settings using complete training sets. Based on the above, it is evident that image generation models alone are ineffective for synthesizing effective segmentation training data.

On the other hand, a handful of works (Zhang et al., 2023; Zhao et al., 2023a) have demonstrated that high-quality synthetic images can be generated by conditioning text-to-image models on dense input maps (*e.g.,* segmentation masks or canny edge). Moreover, these dense-input conditional models provide a strong alignment between the dense input label maps and generated images. These two qualities motivate us to leverage the powerful capabilities of such models for effective segmentation data generation. To this end, we propose a novel segmentation data generation method, coined as **SegGen**, for generating high-quality segmentation training data. As each training sample of image segmentation consists of two components: segmentation masks and the corresponding image, we develop two novel data generation approaches, emphasizing improvements in two distinct aspects of data diversity: (i) segmentation masks and (ii) images. The first data generation approach, named **MaskSyn**, centers around the generation of new segmentation masks. It learns a text-to-mask (Text2Mask) generation model to produce completely new segmentation masks given text prompts. Then, it learns a mask-to-image (Mask2Img) generation model to synthesize images that align with the synthetic segmentation masks. The second data generation approach, named **ImgSyn**, utilizes the above-mentioned Mask2Img model to synthesize new images given human-annotated segmentation masks. With MaskSyn and ImgSyn, we can readily generate a vast array of diverse and high-quality synthetic training data. The combined synthetic data is used to train segmentation models in conjunction with real training samples from human-annotated datasets. Experiments find that our synthetic data can significantly boost the performance of the image segmentation models on challenging benchmarks including ADE20K semantic segmentation, COCO panoptic segmenta-

tion, and COCO instance segmentation, achieving new state-of-the-art performances without using extra human-annotated data. Notably, our method boosts the mIoU of Mask2Former by +2.7 (R50) and +1.3 (Swin-L) on the ADE20K semantic segmentation benchmark. Furthermore, segmentation models trained on our synthetic data exhibit a remarkably stronger ability to generalize across unfamiliar image domains. These solid results confirm the robust efficacy of the proposed SegGen.

We summarize the contribution of this work in three points: **(i)** We propose an innovative generation framework capable of producing high-quality segmentation training data at scale, thus enabling the training of more powerful image segmentation models. **(ii)** We introduce two effective generative models, one for text-to-mask generation and the other for mask-to-image generation. Based on these models, we propose two novel segmentation training data generation approaches, namely MaskSyn and ImgSyn. They significantly improve the data diversity, with MaskSyn focusing on new segmentation masks and ImgSyn on new images. **(iii)** SegGen successfully improves the performance of the leading-edge segmentation models across the highly competitive benchmarks on ADE20K and COCO. Moreover, SegGen enhances the generalization ability of segmentation models towards unseen image domains. Rigorous experiments, including ablation study and peer comparison, strongly suggest the effectiveness of the proposed method.

## 2 RELATED WORK

**Generation for Segmentation** Image segmentation is one of the most studied visual perception problems (Li et al., 2023c). In recent years, there has been a surge in efforts to harness the capabilities of generative models for segmentation tasks. These efforts can be broadly classified into three categories based on their methodologies: (i) Extracting visual features from generative models for segmentation (Baranchuk et al., 2022; Xu et al., 2023; PNVR et al., 2023; Zhao et al., 2023b). These methods harvest the strong representation ability of diffusion models trained on large-scale datasets but are limited in the precision of predicted masks. (ii) Formulating segmentation tasks directly as generative models (Chen et al., 2022; Ji et al., 2023; Wang et al., 2023). Although these methods propose exciting new model architectures, they usually consume higher computational costs while showing unimproved performance compared with conventional segmentation models. (iii) Synthesizing segmentation training data using generative models (Zhang et al., 2021b; Li et al., 2022; Wu et al., 2023b; Li et al., 2023e; Wu et al., 2023a; Xie et al., 2023; Nguyen, 2023). While these methods have showcased commendable results against their respective baselines especially when the training data is highly limited, they have yet to exhibit notably superior performance on the most rigorous benchmarks including ADE20K (150 categories) (Zhou et al., 2017) and COCO (133 categories) (Lin et al., 2014) under fully-supervised setting. A primary concern with these techniques is the subpar quality of the generated segmentation masks. This stems from their dependence on the noisy intermediate features of image generation models during the synthetic mask generation process. In contrast, our data generation pipeline employs a unique and more reasonable workflow: we generate segmentation masks from text, and synthesize images based on segmentation masks. From our experimental findings, our method strongly enhances the existing state-of-the-art segmentation models on different challenging benchmarks.

**Conditional Image Synthesis** Within the realm of conditional image synthesis, Generative Adversarial Networks (Goodfellow et al., 2020; Brock et al., 2019; Karras et al., 2019; Kang et al., 2023; Sauer et al., 2023), Variational Autoencoders (Kingma & Welling, 2013), and Diffusion Models (Sohl-Dickstein et al., 2015; Dhariwal & Nichol, 2021; Saharia et al., 2022; Ho et al., 2020) have been at the forefront. Recently, the open-source research community has shown a burgeoning interest in the latent diffusion-based Stable Diffusion (SD) series for text-to-image synthesis (Rombach et al., 2022). The most recent iteration, SDXL, introduced by Podell et al. (2023), expands the model capacity, yielding significantly enhanced results. Therefore, we build SegGen upon SDXL.

Regarding mask-to-image generation models (Huang et al., 2023; Zeng et al., 2023; Li et al., 2023d), ControlNet (Zhang et al., 2023) and T2I-Adapter (Mou et al., 2023) suggest freezing the parameters of the SD model and introducing a set of more compact, learnable modules. This approach enables image generation conditioned on the given segmentation masks. We adopt the structure of ControlNet in our mask-to-image generation model. Zheng et al. (2023) propose a method for synthesizing novel images based on remote sensing change events. To the best of our knowledge, there have been no prior endeavors on text-to-mask generation.

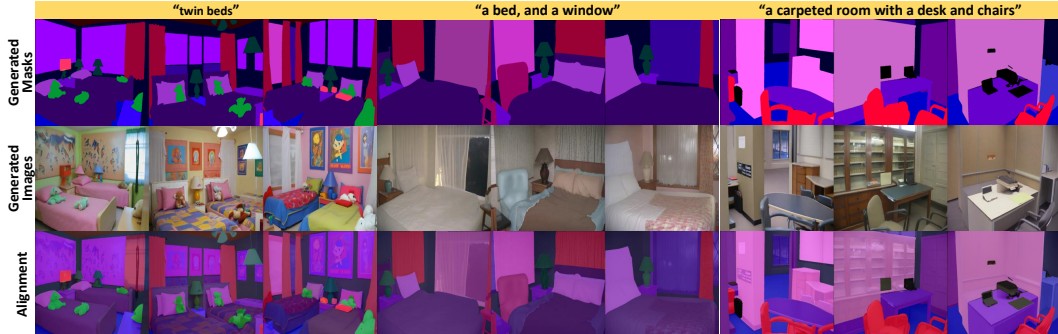

Figure 3: **Generated Samples by MaskSyn on ADE20K:** The third row overlays the mask and the image together to demonstrate the alignment between them. The generated segmentation masks and images demonstrate high perceptual quality and excellent alignment (see more samples in Fig. 13).

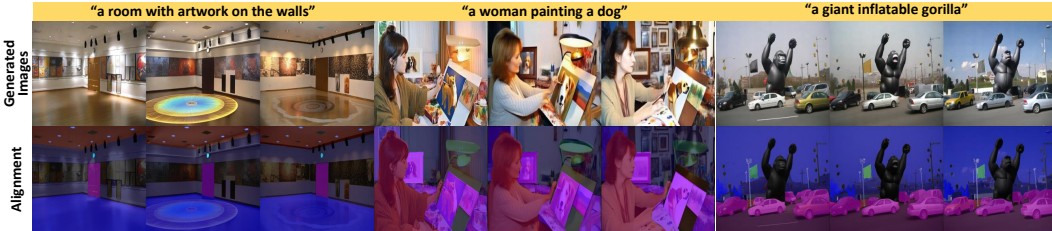

Figure 4: **Generated Samples by ImgSyn on ADE20K:** The generated images exhibit remarkable realism and align well with the human-labeled masks and text prompts (see more samples in Fig. 14).

## 3 METHOD

### 3.1 OVERVIEW

SegGen is designed to synthesize high-quality training samples for improving the performance of segmentation models. The overall workflow is shown on the left of Fig. 2. We first train SegGen with human-annotated training samples from public datasets. After training, we use SegGen to produce new segmentation training samples at scale. The generated training samples are incorporated into the training process of segmentation models with the aim of enhancing the model performance.

### 3.2 MODELS IN SEGGEN

As shown in Fig. 2, we first utilize a captioner model to extract captions of the real training images as text prompts from the target dataset. The text prompts will condition the data generation process. Then, two conditional generative models are introduced: a text-to-mask (Text2Mask) generation model and a mask-to-image (Mask2Img) generation model. Both generative models are built upon the SDXL model (Podell et al., 2023) which provides top-notch image generation quality.

**Captioner Model** To obtain image captions of existing training samples, we employ BLIP2-FlanT5$_{xxl}$ (Li et al., 2023b) model, which is a state-of-the-art vision-language model. We feed the prompt "Question: What are shown in the photo? Answer:" and image as input to the model. The responses serve as text prompts to condition the following generation process.

**Text2Mask Model** We design a Text2Mask model in SegGen for generating diverse segmentation masks based on given text prompts. To leverage the generation capacity of text-to-image generation models pre-trained on large-scale datasets, we encode the segmentation masks (the pixel values are category IDs) as three-channel RGB-like color maps, where one color represents a certain category. From our experiments, the color map reconstructed by VAE (Rombach et al., 2022) in SDXL appears almost indistinguishable from the original input as shown in Fig. 12. Therefore, we can directly fine-tune the text-to-image SDXL-base model with [text, segmentation color map] training pairs, which are from the public image segmentation dataset (*e.g.* ADE20K). During sampling, our Text2Mask model can generate diverse color maps conditioned on text prompts, which are subsequently converted into segmentation masks. Formally, suppose the input text prompt is $\mathbf{T}$, the target height and width are $H$ and $W$, the synthesized color map is $\mathbf{C}_{syn} \in \mathbb{W}^{H \times W \times 3}$, and the synthesized segmentation map (with $N$ masks) is $\mathbf{M}_{syn} \in \mathbb{W}^{H \times W \times N}$, the generation process is formulated as:

$$\begin{aligned} \mathbf{C}_{syn} &= \text{Text2Mask}(\mathbf{T}), \\ \mathbf{M}_{syn} &= f_{color \rightarrow mask}(\mathbf{C}_{syn}), \end{aligned} \quad (1)$$

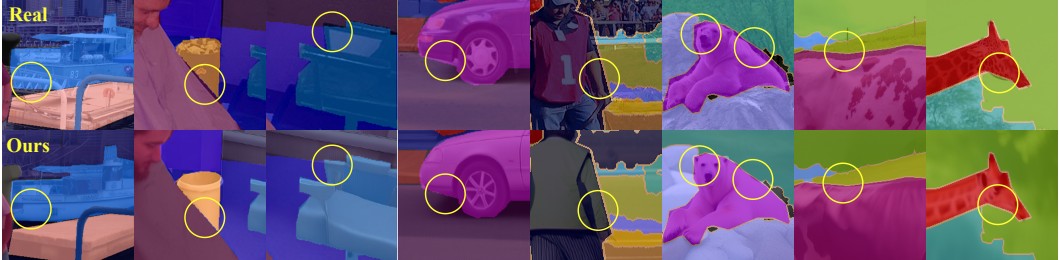

Figure 5: **Zoom-in Comparison of Real Images and Our ImgSyn Images:** As highlighted in the circles, our synthetic images align better with the human-labeled masks in many cases because of the inaccuracies in annotations. The left 4 columns are from ADE20K and the right 4 from COCO.

where $f_{\text{color}\to\text{mask}} : \mathbb{W}^{H\times W\times 3} \to \mathbb{W}^{H\times W\times N}$ is the function that projects the color maps to segmentation masks as detailed in Appendix A.2.

**Mask2Img Model** The goal of the Mask2Img model is to synthesize new images that align well with the given segmentation masks and text prompts. Specifically we adopt the structure of Control-Net (Zhang et al., 2023): we freeze the pre-trained weights of the SDXL-base model and train an additional side network for mask-conditioned image generation. It simultaneously keeps the generalization ability of the pre-trained diffusion model and provides excellent controllable generation ability. The Mask2Img model is trained with the [text, segmentation color map, image] triplets gathered from the training splits of the target datasets. We denote the input segmentation map as $\mathbf{M} \in \mathbb{W}^{H\times W\times N}$, the color map as $\mathbf{C} \in \mathbb{W}^{H\times W\times 3}$, the synthetic image as $\mathbf{I}_{\text{syn}} \in \mathbb{W}^{H\times W\times 3}$, and the generation process is formulated as:

$$\begin{aligned} \mathbf{C} &= f_{\text{mask}\to\text{color}}(\mathbf{M}), \\ \mathbf{I}_{\text{syn}} &= \text{Mask2Img}(\mathbf{T}, \mathbf{C}), \end{aligned} \tag{2}$$

where $f_{\text{mask}\to\text{color}} : \mathbb{W}^{H\times W\times N} \to \mathbb{W}^{H\times W\times 3}$ is the function to convert the segmentation masks into a color map, as explained in Appendix A.2. The segmentation map $\mathbf{M}$ can be human-annotated or synthetic (*i.e.*, $\mathbf{M}_{\text{syn}}$ from Eq. 1).

### 3.3 DATA GENERATION APPROACHES

With the aforementioned generative models, SegGen proposes two approaches for synthesizing new segmentation training samples: MaskSyn and ImgSyn. MaskSyn focuses on enhancing the diversity of synthetic segmentation masks, whereas ImgSyn concentrates on diversifying synthetic images. These generation approaches are illustrated on the right of Fig. 2.

**MaskSyn** MaskSyn starts with a real training sample pair [image, segmentation masks] from a human-annotated segmentation dataset. It first extracts the caption of the image with the image captioner model. The caption serves as a text prompt and is used to generate a set of diverse new segmentation masks with the Text2Mask model following Eq. 1. Subsequently, the new segmentation masks and the corresponding text prompt are fed into the Mask2Img model to generate a new image that aligns well with its mask. As such, each training sample crafted by MaskSyn includes both a novel segmentation mask and a new image. MaskSyn effectively increases the data diversity in segmentation masks to a great extent. Some generated samples are shown in Fig. 3.

**ImgSyn** Different from MaskSyn, ImgSyn focuses on increasing the data diversity of images based on human-annotated segmentation masks. For each real training sample pair [image, segmentation mask], it takes the human-annotated segmentation mask and the text prompt extracted from the image as input, and generates a set of varied images that align well with the human-annotated mask. In this way, new training samples consisting of human-labeled masks and new synthetic images are generated. ImgSyn can also be viewed as a kind of data augmentation method that enhances the data diversity on the image side. Our experiments reveal a remarkably high alignment between the synthetic images and their respective segmentation masks. Some synthesized results are showcased in Fig. 4. Our machine-generated synthetic images achieve a better mask-image alignment than real images in many cases, as shown in Fig. 5. This phenomenon occurs because human annotations tend to be imperfect due to the high difficulty of annotating segmentation masks.

| Method | Venue | Backbone | Crop Size | Iterations | mIoU (s.s.) | mIoU (m.s.) |
|---|---|---|---|---|---|---|
| MaskFormer (Cheng et al., 2021) | NeurIPS 2021 | R50 | 512 | 160k | 44.5 | 46.7 |
| Mask DINO (Li et al., 2023a) | CVPR 2023 | R50 | 512 | 160k | 48.7 | - |
| OneFormer (Jain et al., 2023) | CVPR 2023 | R50 | 512 | 160k | 47.3 | - |
| Mask2Former (Cheng et al., 2022) | CVPR 2022 | R50 | 512 | 160k | 47.2 | 49.2 |
| Mask2Former w/ **SegGen (ours)** | - | R50 | 512 | 160k | **49.9 (+2.7)** | **51.4 (+2.2)** |
| MaskFormer (Cheng et al., 2021) | NeurIPS 2021 | Swin-L | 640 | 160k | 54.1 | 55.6 |
| Mask DINO (Li et al., 2023a) | CVPR 2023 | Swin-L | 640 | 160k | 56.6 | - |
| OneFormer (Jain et al., 2023) | CVPR 2023 | Swin-L | 640 | 160k | 57.0 | 57.7 |
| Mask2Former (Cheng et al., 2022) | CVPR 2022 | Swin-L | 640 | 160k | 56.1 | 57.3 |
| Mask2Former w/ **SegGen (ours)** | - | Swin-L | 640 | 160k | **57.4 (+1.3)** | **58.7 (+1.4)** |

Table 1: **Semantic segmentation on ADE20K `val` (150 categories):** Synthetic data is used as a data augmentation. With the same model structures and training recipes, our SegGen boosts the performance of Mask2Former by a large margin and establishes a new SOTA performance without extra real data under both single-scale (s.s.) and multi-scale (m.s.) test settings.

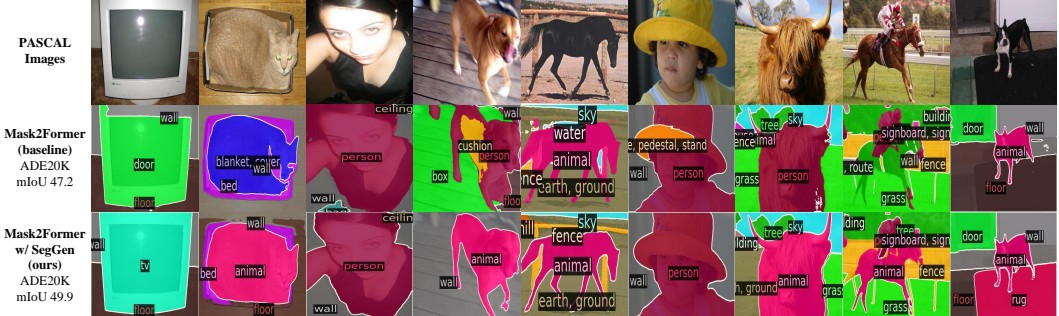

Figure 6: **Segmentation Generalization Ability**: Segmentation outputs on unseen domain (images from PASCAL (Everingham et al., 2015)). The models are trained on ADE20K. Training with our SegGen demonstrates enhanced robustness towards unfamiliar domains.

### 3.4 TRAINING SEGMENTATION MODELS WITH SYNTHETIC DATA

The final goal of this work is to improve the performance of current segmentation models with the synthetically generated training data. Therefore, we use both synthetic data produced by SegGen and training data from existing datasets in the training process of segmentation models.

We empirically investigate two training strategies utilizing the synthetic data: (i) Synthetic data augmentation strategy. The synthetic data is used for random data augmentation. In every iteration in the training process, each real training sample is replaced by the synthetic training sample with a probability $p_{aug}$. (ii) Synthetic data pre-training strategy. It comprises two training stages: pre-training and fine-tuning. In the pre-training stage, we pre-train the segmentation models on synthetic data, so that they learn good weights that are transferable and favorable for fine-tuning. In the subsequent fine-tuning stage, the segmentation models are trained with solely human-annotated data.

## 4 EXPERIMENTS

To accurately evaluate the effectiveness of our data generation method in improving segmentation performance, we adopt mainstream segmentation models and commonly used evaluation benchmarks for several typical segmentation tasks. The experiments are conducted mostly under the fully-supervised learning setting, meaning all human-annotated training samples from the evaluated datasets are used alongside our synthetic data. To guarantee an unbiased assessment of the impact of our synthetic data, we keep the architectures of the segmentation models and training protocols consistent with their respective original implementations throughout our studies.

### 4.1 IMPLEMENTATION DETAILS

**Segmentation Datasets and Evaluation** We conduct experiments on three image segmentation benchmarks following the main experimental settings of Mask2Former (Cheng et al., 2022): ADE20K semantic segmentation (Zhou et al., 2016), COCO panoptic segmentation, and COCO instance segmentation (Lin et al., 2014). Our evaluation uses all 150 classes for ADE20K and 133 classes for COCO. We use all the images from the training splits in the training of segmentation models. For semantic segmentation, we show the mean Intersection-over-Union metric (mIoU). For

| Method | Backbone | Queries | Epochs | PQ | PQ$^{Th}$ | PQ$^{St}$ | AP$^{Th}_{pan}$ | mIoU$_{pan}$ |
|---|---|---|---|---|---|---|---|---|
| DETR (Carion et al., 2020) | R50 | 100 | 500+25 | 43.4 | 48.2 | 36.3 | 31.1 | - |
| MaskFormer (Cheng et al., 2021) | R50 | 100 | 300 | 46.5 | 51.0 | 39.8 | 33.0 | 57.8 |
| Mask2Former (Cheng et al., 2022) | R50 | 100 | 50 | 51.9 | 57.7 | 43.0 | 41.7 | 61.7 |
| Mask DINO (Li et al., 2023a) | R50 | 300 | 50 | 53.0 | 59.1 | 43.9 | 43.3 | - |
| Mask2Former (Cheng et al., 2022) | R50 | 100 | 50+50 | 52.0 | 57.9 | 43.4 | 42.0 | 61.0 |
| Mask2Former w/ **SegGen (ours)** | R50 | 100 | 50+50 | **52.7** (+0.7) | **58.8** (+0.9) | **43.6** (+0.2) | **43.1** (+1.1) | **62.6** (+1.6) |
| Mask DINO (Li et al., 2023a) | R50 | 300 | 50+50 | 53.4 | 59.3 | 44.4 | 44.2 | 60.5 |
| Mask DINO w/ **SegGen** (ours) | R50 | 300 | 50+50 | **54.0** (+0.6) | **60.2** (+0.9) | **44.7** (+0.3) | **45.4** (+1.2) | **61.5** (+1.0) |
| MaskFormer (Cheng et al., 2021) | Swin-L | 100 | 300 | 52.7 | 58.5 | 44.0 | 40.1 | 64.8 |
| OneFormer (Jain et al., 2023) | Swin-L | 150 | 100 | 57.9 | 64.4 | 48.0 | 49.0 | 67.4 |
| Mask2Former (Cheng et al., 2022) | Swin-L | 200 | 100 | 57.8 | 64.2 | 48.1 | 48.6 | 67.4 |
| Mask DINO | Swin-L | 300 | 50 | 58.3 | 65.1 | 48.0 | 50.6 | - |
| Mask2Former (Cheng et al., 2022) | Swin-L | 200 | 100+100 | 57.3 | 64.3 | 46.8 | 48.0 | 66.2 |
| Mask2Former w/ **SegGen (ours)** | Swin-L | 200 | 100+100 | **58.0** (+0.7) | **64.5** (+0.2) | **48.1** (+1.3) | **48.8** (+0.8) | **67.4** (+1.2) |
| Mask DINO (Li et al., 2023a) | Swin-L | 300 | 50+50 | 58.6 | 65.4 | 48.3 | 50.4 | 67.0 |
| Mask DINO w/ **SegGen** (ours) | Swin-L | 300 | 50+50 | **59.3** (+0.7) | **65.9** (+0.5) | **49.3** (+1.0) | **51.1** (+0.7) | **68.1** (+1.1) |

Table 2: **Panoptic segmentation on COCO panoptic `val2017` (133 categories):** Synthetic data is used for pre-training. Our SegGen significantly improves the performance compared with real data pre-training baselines, and establishes new SOTA performance without extra real data.

instance segmentation, the average precision (AP) is used. For panoptic segmentation, we report panoptic quality (PQ), "thing" instance segmentation AP$^{Th}_{pan}$, and semantic segmentation mIoU$_{pan}$.

**Segmentation Models** We adopt Mask2Former (Cheng et al., 2022), a recently-proposed prevalent transformer model, as the default segmentation model. Two typical backbones, *i.e.*, R50 (He et al., 2016) and Swin-L (Liu et al., 2021), are studied. We keep the official implementation and training hyper-parameters of the segmentation models unchanged. For more training details please refer to their paper. We also conduct experiments on Mask DINO (Li et al., 2023a), a detection-aided segmentation model, and HRNet W48 (Wang et al., 2019), a representative fully-convolutional model. More implementation details are introduced in Appendix A.3.

**Data Generation** The training of generative models is detailed in Appendix A.4. During data sampling, for each training sample in the ADE20K semantic segmentation dataset, we produce 10 synthetic mask-image pairs using MaskSyn, resulting in 202,100 training samples. Additionally, we synthesize 50 images based on each human-labeled mask with ImgSyn, leading to a total of 1,010,500 samples. When it comes to COCO instance and panoptic segmentation, obtaining instance information from color maps is challenging, as explained in Appendix A.2. Hence, we solely rely on ImgSyn for COCO data synthesis. By generating 10 synthetic images conditioned on each human-labeled panoptic segmentation mask via ImgSyn, our synthetic set amounts to 1,182,870 synthetic samples, which are used in the training of both panoptic and instance segmentation models.

## 4.2 MAIN RESULTS

**ADE20K Semantic Segmentation** We employ the synthetic data augmentation strategy with $p_{aug} = 60\%$ for Mask2Former, and show the results in Table 1. SegGen significantly boosts the mIoU of Mask2Former R50 by **+2.7** for single-scale inference and **+2.2** for multi-scale inference, achieving 49.9 and 51.4 correspondingly. The Swin-L variant is also largely improved from 56.1/57.3 (single-scale/multi-scale) to 57.4/58.7 (**+1.3/+1.4**). Notably, our method helps Mask2Former surpass the newer methods such as Mask DINO and OneFormer (Jain et al., 2023), while establishing new SOTA results for R50 and Swin-L settings without using additional human-annotated data.

**COCO Panoptic Segmentation** On COCO we adopt the synthetic data pre-training strategy to utilize our synthetic data. Specifically, we pre-train the models purely on our synthetic data and fine-tune the models on real data. The results are demonstrated in Table 2. For a fair comparison, we compare with models pre-trained on purely real data with the same training settings. Compared with the real data pre-trained baselines, SegGen consistently brings performance gains on all metrics with both R50 and Swin-L backbones, achieving new state-of-the-art performance without extra real data. Notably, for Mask DINO with Swin-L backbone, PQ is improved by +0.7, "thing" instance segmentation AP$^{Th}_{pan}$ is increased by +0.7, and semantic segmentation mIoU$_{pan}$ is boosted by +1.1. These noteworthy findings demonstrate that training with our synthetic data systematically improves performance across various segmentation tasks. We observe similar significant improvement on Mask2Former, further showing the effectiveness of our synthetic data on different models.

**COCO Instance Segmentation** The performance on instance segmentation, both before and after incorporating our SegGen synthetic data, is detailed in Table 3. Compared to the baselines

| Method | Backbone | Queries | Epochs | AP | AP$^S$ | AP$^M$ | AP$^L$ |
|---|---|---|---|---|---|---|---|
| Mask R-CNN (He et al., 2017) | R50 | anchors | 400 | 42.5 | 23.8 | 45.0 | 60.0 |
| HTC (Chen et al., 2019) | R50 | anchors | 36 | 39.7 | 22.6 | 42.2 | 50.6 |
| QueryInst (Fang et al., 2021) | R50 | 300 | 36 | 40.6 | 23.4 | 42.5 | 52.8 |
| MaskFormer (Cheng et al., 2021) | R50 | 100 | 300 | 34.0 | 16.4 | 37.8 | 54.2 |
| Mask2Former (Cheng et al., 2022) | R50 | 100 | 50 | 43.7 | 23.4 | 47.2 | 64.8 |
| Mask2Former (Cheng et al., 2022) | R50 | 100 | 50+50 | 44.1 | 23.4 | 47.7 | 66.1 |
| Mask2Former w/ **SegGen** (ours) | R50 | 100 | 50+50 | **44.6** (+0.5) | **24.0** (+0.6) | **48.2** (+0.5) | **66.2** (+0.1) |
| Swin-HTC++ (Liu et al., 2021) | Swin-L | anchors | 72 | 49.5 | 31.0 | 52.4 | 67.2 |
| QueryInst (Fang et al., 2021) | Swin-L | 300 | 50 | 48.9 | 30.8 | 52.6 | 68.3 |
| Oneformer (Jain et al., 2023) | Swin-L | 150 | 100 | 48.9 | - | - | - |
| Mask2Former (Cheng et al., 2022) | Swin-L | 200 | 100 | 50.1 | 29.9 | 53.9 | 72.1 |
| Mask2Former (Cheng et al., 2022) | Swin-L | 200 | 100+100 | 49.5 | 29.2 | 53.8 | 70.5 |
| Mask2Former w/ **SegGen** (ours) | Swin-L | 200 | 100+100 | **50.3** (+0.8) | **31.2** (+2.0) | **54.3** (+0.5) | **72.2** (+1.7) |

Table 3: **Instance segmentation on COCO `val2017` (80 categories):** Synthetic data is used for pre-training. Our SegGen significantly improves the performance of instance segmentation compared with real data pre-training baselines.

| Method | Iterations | Real Samples | mIoU |
|---|---|---|---|
| Baseline | 160K | 20210 | 47.2 |
| — w/ **SegGen** MaskSyn | 160K | 20210 | 48.5 |
| — w/ **SegGen** ImgSyn | 160K | 20210 | 49.3 |
| — w/ **SegGen** ImgSyn + MaskSyn | 160K | 20210 | **49.9** |
| Baseline | 160K | 1000 | 23.7 |
| — w/ **SegGen** MaskSyn | 160K | 1000 | 26.5 |
| — w/ **SegGen** ImgSyn | 160K | 1000 | 24.1 |
| — w/ **SegGen** ImgSyn + MaskSyn | 160K | 1000 | **27.1** |

(a) Mask2Former R50

| Method | Iterations | Real Samples | mIoU |
|---|---|---|---|
| Baseline | 100K | 20210 | 44.4 |
| — w/ **SegGen** MaskSyn | 100K | 20210 | 44.8 |
| — w/ **SegGen** ImgSyn | 100K | 20210 | 45.4 |
| — w/ **SegGen** ImgSyn + MaskSyn | 100K | 20210 | **45.6** |
| Baseline | 100K | 1000 | 23.6 |
| — w/ **SegGen** MaskSyn | 100K | 1000 | 26.0 |
| — w/ **SegGen** ImgSyn | 100K | 1000 | 23.9 |
| — w/ **SegGen** ImgSyn + MaskSyn | 100K | 1000 | **26.9** |

(b) HRNet W48

Table 4: **Ablation Study of MaskSyn and ImgSyn on ADE20K:** Both MaskSyn and ImgSyn notably enhance the segmentation performance, while MaskSyn has a more pronounced impact when there are fewer real samples available on both Mask2Former and HRNet.

pre-trained on real data, our technique is superior across all metrics. Specifically, there is a +0.5 improvement in AP with the R50 backbone, and a +0.8 increase when using the Swin-L backbone.

**Segmentation Performance on Unseen Domains** We visualize the segmentation results of Mask2Former R50 on images from PASCAL `val` dataset (Everingham et al., 2015) in Fig. 1 and Fig. 6, comparing models trained both with and without our synthetic data. The models are trained on COCO or ADE20K. When trained using our extensively varied synthetic data, the segmentation model demonstrates notably improved performance on the unseen domain, thanks to its stronger generalization ability. We further study the segmentation results on AI-generated images, which are synthesized using different image generation models, in Appendix A.5.

**Visual Analysis of Generated Samples by MaskSyn** We randomly select some samples generated by MaskSyn in Fig. 3 and Fig. 13. The generative models are trained on ADE20K. Given the text prompts, our Text2Mask model can generate diverse segmentation masks, and the Mask2Img model produces realistic synthetic images with a good alignment with the masks and text prompts.

**Visual Analysis of Generated Samples by ImgSyn** We showcase the outputs by ImgSyn in Fig. 4 and Fig. 14. The generated images are in good agreement with the text prompts and human-labeled segmentation masks. Furthermore, we compare the mask-image alignment of the real training samples and our synthetic training samples in Fig. 5. Our synthetic samples exhibit superior alignment quality in many cases, compared to human annotations which are often imperfect on the boundaries.

### 4.3 MORE QUANTITATIVE STUDY

**Ablation Study of MaskSyn and ImgSyn** We evaluate the data generated by MaskSyn and ImgSyn separately and jointly, by examining their impacts on the performance of segmentation models, as shown in Table 4. The experiments are conducted on ADE20K with Mask2Former R50 and HRNet W48. We find that the training samples generated by both MaskSyn and ImgSyn bring significant performance gains compared with the baseline. When combining these two types of synthetic data together, the segmentation model can achieve the best performance. We delve deeper into a scenario where only 1,000 real training samples are available in the training of both generative models and segmentation models. It is observed that MaskSyn substantially enhances the segmentation results in situations with limited real data. This could be attributed to the ability of MaskSyn to amplify data diversity by creating entirely new segmentation masks and images. Moreover, the significant

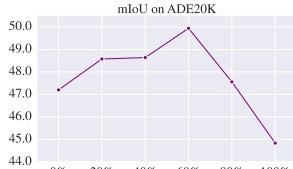

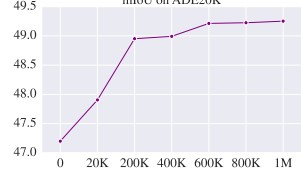

| Method | Bus | Car | Person | Mean |
|---|---|---|---|---|
| Pure Real Data | **87.9** | **82.5** | **79.4** | **83.3** |
| DiffuMask (ICCV 2023) | 43.4 | 67.3 | 60.2 | 57.0 |
| **MaskSyn** | 72.1 | 70.2 | 77.2 | 73.2 |
| **ImgSyn** | 82.3 | 79.1 | 78.64 | 80.0 |
| **MaskSyn+ImgSyn** | 86.4 | 81.5 | 78.7 | 82.2 |

Figure 7: ADE20K mIoU of Mask2Former with different augmentation probability $p_{\text{aug}}$.

Figure 8: ADE20K mIoU of Mask2Former R50 using different numbers of synthetic training samples.

Table 5: **Comparison with DiffuMask (Wu et al., 2023b) on ADE20K:** All models trained purely on synthetic data. Our performance is even comparable to training solely with real data.

| Method | Iterations | mIoU |
|---|---|---|
| Syn. Pre-Train | 160K+160K | 47.2 |
| Syn. Data Aug | 160K | **49.9** |

| Method | Epochs | PQ | AP |
|---|---|---|---|
| Syn. Pre-Training | 50+50 | **52.7** | **44.6** |
| Syn. Data Aug | 50 | 51.3 | 43.2 |

(a) **ADE20K Semantic Segmentation:** Using synthetic data as random data augmentation obtains the best performance on ADE20K.

(b) **COCO Panoptic Segmentation and Instance Segmentation:** Using synthetic data for pre-training achieves better performance on COCO.

Table 6: Different training strategies with synthetic data using Mask2Former R50.

performance improvements achieved by our SegGen, trained on a mere 1,000 real samples, serve as a testament to the data efficiency and robustness of our proposal.

**Ablation Study of Synthetic Data Augmentation Probability $p_{\text{aug}}$ on ADE20K** We investigate the effect of $p_{\text{aug}}$ in Fig. 7. When $p_{\text{aug}} = 0$, it implies training using only real data, whereas $p_{\text{aug}} = 100\%$ means training entirely with synthetic data. The best results are achieved at $p_{\text{aug}} = 60\%$. It is noteworthy that training exclusively on our synthetic data results in an impressive mIoU of 44.8.

**Ablation Study of Synthetic Data Size** To evaluate the impact of synthetic data volume, we use different numbers of synthetic training samples produced by ImgSyn in training and report the results in Fig. 8. $p_{\text{aug}}$ is fixed at 60%. We notice a substantial improvement in performance when increasing the synthetic data quantity from 20K to 200K, underscoring the importance of collecting a significantly larger synthetic dataset. The performance appears to plateau after 600K. More studies on the scale of synthetic data are presented in Appendix A.6.

**Comparison with Related Data Generation Method** In Table 5, we compare various versions of our SegGen with DiffuMask (Wu et al., 2023b), which is a recently-proposed segmentation data generation approach. All methods employ the Mask2Former R50 model and are *purely trained on synthetic samples*. We examine the IoU for three common classes, adhering to the criteria designed in DiffuMask. Our method markedly surpasses DiffuMask and demonstrates performance comparable to training purely with real data, highlighting the superior quality of our synthetic data.

**Ablation Study of Training Strategies** We conduct an ablation study using different training strategies with synthetic data on ADE20K (Table 6a) and COCO (Table 6b). We find that using the synthetic data augmentation strategy achieves better performance on ADE20K semantic segmentation, While on COCO panoptic and instance segmentation, the synthetic pre-training strategy works better. We believe that due to the limited size of the ADE20K training set, which contains merely ∼20k images, segmentation models easily overfit the scarcely-available training images and segmentation layouts. Utilizing synthetic data (including synthetic masks and images) for data augmentation can help mitigate such an overfitting problem. On the other hand, for COCO, given its larger training set size (around 100K images), overfitting is less of a concern. Hence, it is sufficient to provide the COCO models with good initial weights pre-trained on our synthetic data. Meanwhile, previous work (Gupta et al., 2019) finds that the annotation accuracy of COCO is significantly more biased than ADE20K, a finding that aligns with our visualization results depicted in Fig. 5. A domain gap exists between the COCO data and our high-quality synthetic data. Hence, employing our method for pre-training and incorporating real data for fine-tuning on COCO is a more favorable approach.

## 5 CONCLUSION

We present SegGen, a highly-effective data synthesis method for image segmentation. SegGen introduces two data generation approaches, namely MaskSyn and ImgSyn, to generate synthetic segmentation masks and images, with the help of the designed text-to-mask and mask-to-image generation models. Our method not only strongly enhances the model performance on semantic, panoptic, and instance segmentation benchmarks, but also substantially improves segmentation generalization ability. The project will be open-source to encourage further research on image segmentation.

## ETHICS STATEMENT

The proposed method focuses on the improvement of image segmentation with generation models without introducing new tasks or human-annotated datasets. While there has been discussion regarding the ethical implications of generative models (Rostamzadeh et al., 2021), our research does not introduce any additional ethical concerns.

## REPRODUCIBILITY STATEMENT

Details regarding model training and the generation of synthetic data can be found in Section 4.1, Appendix A.4, and Appendix A.3. More importantly, we will make our project publicly available, encompassing the code, model weights, and our synthetic dataset, to assist the community in pursuing similar studies.

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

## A    Appendix

### A.1    Comparison with Previous Segmentation Data Generation Methods

We delve deeper into the distinctions between our workflow and the previous ones, such as Dif-fuMask (Wu et al., 2023b), Grounded Diffusion (Li et al., 2023e) and DatasetGAN (Zhang et al., 2021b), as shown in Fig. 9. These methods rely on the intermediate features of image generation models for producing segmentation masks, either by using a small-scale segmentation module as in DatasetGAN and Grounded Diffusion, or by conducting unstable post-processing as in DiffuMask. The dependence on the visibly noisy features from image generation models, which are not meant for segmentation tasks, leads to low quality in the synthetic masks. In contrast, our SegGen stands out with its unique approach by directly synthesizing new segmentation masks from text prompts using the proposed Text2Mask generation model, and then synthesizing images conditioned on the synthetic segmentation masks (MaskSyn) or human-labeled segmentation masks (ImgSyn). Our workflow fully leverages the powerful capacity of conditional generative models (Rombach et al., 2022; Zhang et al., 2023).

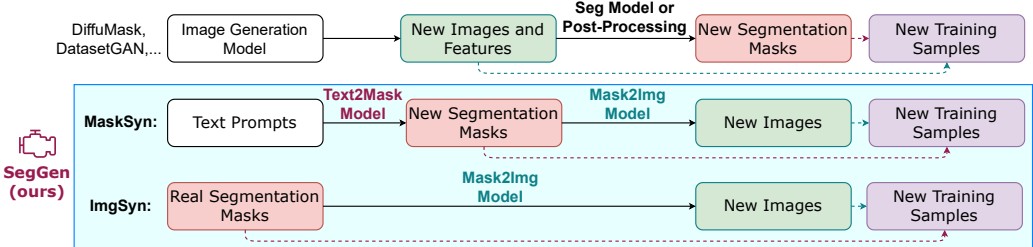

Figure 9: **Illustration of SegGen**: Earlier methods for generating segmentation data rely on inter-mediate features of image generation models (ImgGen models), resulting in subpar mask quality. Our SegGen adopts a more reasonable approach: we first create new masks from text prompts via a proposed Text2Mask generation model and then synthesize images from the synthetic/human-labeled segmentation masks via a Mask2Img generation model. This methodology fully utilizes the powerful data generation ability of conditional generative models.

### A.2    Conversion between Segmentation Masks and Color Maps

**Projection from Segmentation Masks to Color Maps** $f_{\text{mask}\rightarrow\text{color}}$. For semantic segmentation, the value of each pixel on the segmentation mask corresponds to a category ID, allowing us to convert the masks directly into an RGB color map using a pre-defined lookup table. For panoptic and instance segmentation, after mapping the category IDs to color maps, it is essential to outline each segment with a special edge color on the color map. This ensures the model recognizes the specific instance it belongs to.

**Projection from Color Maps to Segmentation Masks** $f_{\text{color}\rightarrow\text{mask}}$. For semantic segmentation, projecting color maps to segmentation masks is straightforward. For each pixel on the color maps, we identify its nearest color (with Euclidean distance) in the aforementioned lookup table, and assign the corresponding class to the pixel in the segmentation masks. However, it is difficult to discern instance membership from the RGB color maps. Consequently, the design of $f_{\text{color}\rightarrow\text{mask}}$ for instance and panoptic segmentation remains an open question.

### A.3    More Implementation Details about Segmentation Models

**Mask2Former** We follow the official training scripts and settings from their GitHub repository. The initial learning rate is $10^{-4}$, weight decay rate is $5 \times 10^{-2}$, and the batch size is 16. For more implementation details please refer to Cheng et al. (2022).

**Mask DINO** Mask DINO is a recently proposed segmentation transformer aided by object detection networks. We use the official training scripts available on their GitHub repository and adhere to their training configurations. The starting learning rate is set at $10^{-4}$, the weight decay rate at $5 \times 10^{-2}$,

and a batch size of 16. For specifics on the loss structure and further implementation details, kindly consult Li et al. (2023a).

**HRNet** For experiments with HRNet (Wang et al., 2019), we use the W48 version, and train it for 100K iterations. SGD optimizer is used with a starting learning rate of $2 \times 10^{-2}$. A polynomial learning rate scheduler is used with a power parameter set to 0.9. The weight decay rate is $10^{-4}$. Random flipping data augmentation is incorporated.

## A.4 MORE IMPLEMENTATION DETAILS ABOUT GENERATIVE MODELS

The Text2Mask and Mask2Img models are both based on the SDXL-base model (Podell et al., 2023). They are trained on the training splits of the target segmentation datasets (*i.e.* ADE20K or COCO) separately for 30,000 iterations using a learning rate of $10^{-5}$. AdamW optimizer is employed, and the models are trained at a resolution of 768. Pre-trained weights from SDXL-base (Podell et al., 2023) are utilized. Random flipping data augmentation is used. During sampling, the default EDM sampler (Karras et al., 2022) is used. The sampling steps are 200 in the Text2Mask model and 40 in the Mask2Img model.

## A.5 SEGMENTATION GENERALIZATION ABILITY ON AI-GENERATED IMAGES

We randomly generate a set of high-quality images with the latest text-to-image generation methods: Kandinsky 2 by Forever (2023), IF by DeepFloyd (2023), and DALL-E by OpenAI (2021). We visualize the segmentation results of Mask2Former R50 trained both with and without our synthetic data in Fig. 1 and Fig. 10. The models are trained on COCO or ADE20K. When trained using our extensively varied synthetic data, the segmentation model demonstrates notably improved performance on the unseen images synthesized by generative models, demonstrating its stronger generalization ability.

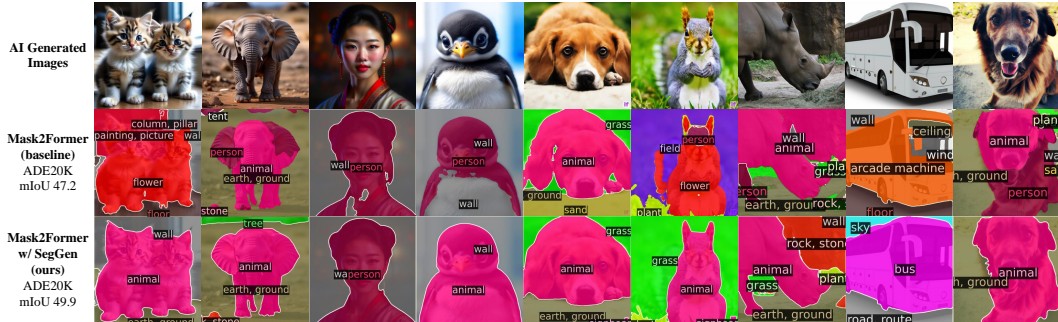

Figure 10: **Segmentation Generalization Ability** : Segmentation outputs of Mask2Former before and after training with our SegGen on ADE20K. The first 3 images are generated by Kandinsky 2, the middle 3 by IF, and the later 3 by DALL-E. Training with our synthetic data brings significantly stronger generalization ability to the segmentation model.

## A.6 MORE STUDY ON SYNTHETIC DATA SCALE

To better understand the impact of synthetic data volume, we train the Mask2Former R50 model using different quantities of ImgSyn samples (*i.e.* 20K and 1M), and show the training loss and validation mIoU in Fig. 11. The synthetic data augmentation probability remains constant at 60%.

We find that utilizing 20K synthetic samples results in a lower training loss compared to using 1M samples; however, the validation mIoU is significantly better with the larger synthetic dataset. Statistically, when we use only 20K synthetic samples, the model encounters each synthetic sample approximately 76.8 times during training. In contrast, with 1M synthetic samples, each sample is presented to the model about 1.5 times. This indicates that when the synthetic dataset is not large enough, there is a risk of the model overfitting to the synthetic data. Hence, employing a more extensive synthetic dataset is very beneficial for training segmentation models, as it prevents the model from easily memorizing all the synthetic training samples. Our SegGen can easily scale up the synthetic data by sampling new masks and images using the proposed generative models.

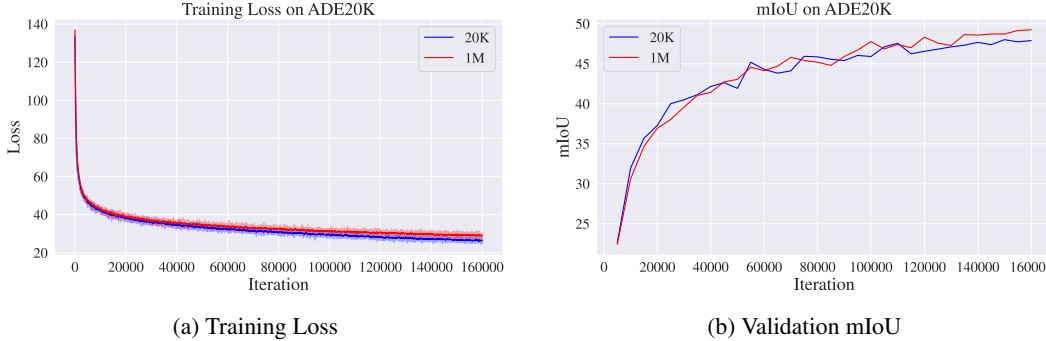

(a) Training Loss

(b) Validation mIoU

Figure 11: **Training loss and validation mIoU on ADE20K:** Using a limited amount of synthetic data can result in overfitting and subsequently worse performance. Therefore we generate more than 1M synthetic training samples, which significantly mitigates the issue. This experiment highlights the importance of the scale of synthetic data.

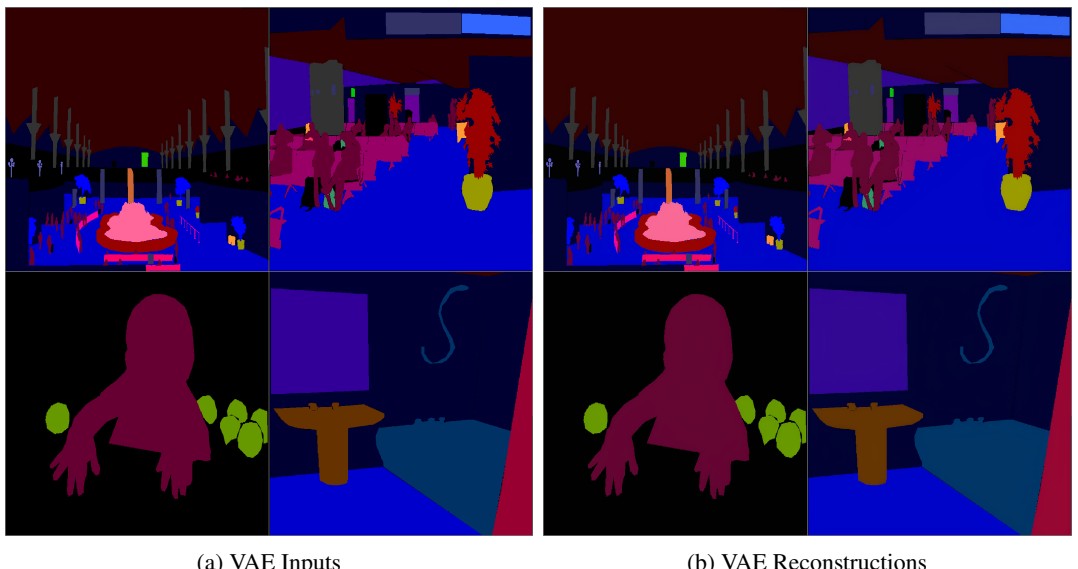

(a) VAE Inputs

(b) VAE Reconstructions

Figure 12: **Visual Analysis of Color Map Reconstruction Using VAE:** The pre-trained VAE of SDXL can effectively reconstruct color maps, establishing the foundation of our Text2Mask model.

## A.7 VISUAL ANALYSIS OF COLOR MAP RECONSTRUCTION CAPABILITY OF VAE

We show the inputs and reconstruction outcomes of variational autoencoders (VAE) in our Text2Mask model in Fig. 12. The VAE, which is borrowed from SDXL (Podell et al., 2023), demonstrates an impressive ability to reconstruct the color maps, enabling our training of the Text2Mask generation model.

## A.8 VISUAL ANALYSIS OF GENERATED SAMPLES

In Fig.13 and Fig.14, we present additional samples generated by our MaskSyn and ImgSyn on the ADE20K dataset. Fig. 15 displays the generated samples on the COCO dataset. These illustrations highlight the superior quality of the synthetic masks and images produced by our SegGen.

## A.9 ABLATION STUDY OF DATA AUGMENTATION TECHNIQUES

In order to verify the efficacy of our approach without employing conventional data augmentation techniques, we carry out an experiment where we eliminate color augmentation, random flipping,

and random cropping from Mask2Former R50. We train it using our SegGen data on ADE20K. The outcomes of this experiment are presented in Table 7. Our observations demonstrate that in the absence of data augmentation methods, our synthetic data has a more pronounced impact on enhancing the segmentation performance. Notably, it brings a notable increase in mIoU by +4.8.

| Method | Iterations | Data Aug | mIoU |
|---|---|---|---|
| Baseline | 160K | Yes | 47.2 |
| — w/ **SegGen** | 160K | Yes | **49.9 (+2.7)** |
| Baseline | 160K | No | 41.4 |
| — w/ **SegGen** | 160K | No | **46.2 (+4.8)** |

Table 7: Ablation study of removing regular data augmentation methods. When these data augmentation methods are unavailable, our SegGen brings a more significant improvement.

## A.10    ABLATION STUDY OF DIFFERENT NUMBERS OF MASKSYN SAMPLES

We conduct a series of experiments to examine the influence of using different numbers of MaskSyn samples, as shown in Table 8. We use Mask2Former R50 as the segmentation model. Similar to the observations in Figure 8 for ImgSyn, we achieve significantly improved performance after augmenting the quantity of MaskSyn samples. This further underscores the importance of generating large-scale segmentation data to train more effective segmentation models.

| Method | Iterations | MaskSyn Samples | mIoU |
|---|---|---|---|
| Baseline | 160K | 0 | 47.2 |
| **SegGen** MaskSyn | 160K | 40K | 47.3 |
| **SegGen** MaskSyn | 160K | 80K | 47.6 |
| **SegGen** MaskSyn | 160K | 160K | 48.1 |
| **SegGen** MaskSyn | 160K | 200K | 48.5 |

Table 8: Ablation study of using different numbers of MaskSyn samples in training Mask2Former R50. More synthetic samples leads to better performance.

## A.11    ADDITIONAL EXPERIMENTS USING STABLE DIFFUSION V1.5

We conduct an extended experiment on ADE20K to use Stable Diffusion v1.5 (SD-1.5) as the base model in our text-to-mask and mask-to-image generation models. We synthesize 200K MaskSyn samples and 200K ImgSyn samples. We find that the generated samples maintain a similarly high quality compared to the ones generated by our previous models using SDXL as base model. We then use the newly generated data to train Mask2Former R50 on ADE20K alongside the real data, and show the results in the table below. Remarkably, our SegGen shows a significant performance gain of +2.2 mIoU. The performance improvement is close to our SDXL version (+2.7 mIoU), which demonstrates the strong robustness of our SegGen framework.

| Method | Iterations | mIoU |
|---|---|---|
| Baseline | 160K | 47.2 |
| — w/ **SegGen** MaskSyn | 160K | 48.3 |
| — w/ **SegGen** ImgSyn | 160K | 48.9 |
| — w/ **SegGen** MaskSyn+ImgSyn | 160K | **49.4** |

Table 9: Additional Experiments using Stable Diffusion v1.5. Our SegGen still brings significant performance improvement even when using a weaker generation base model, which demonstrates the robustness of SegGen.

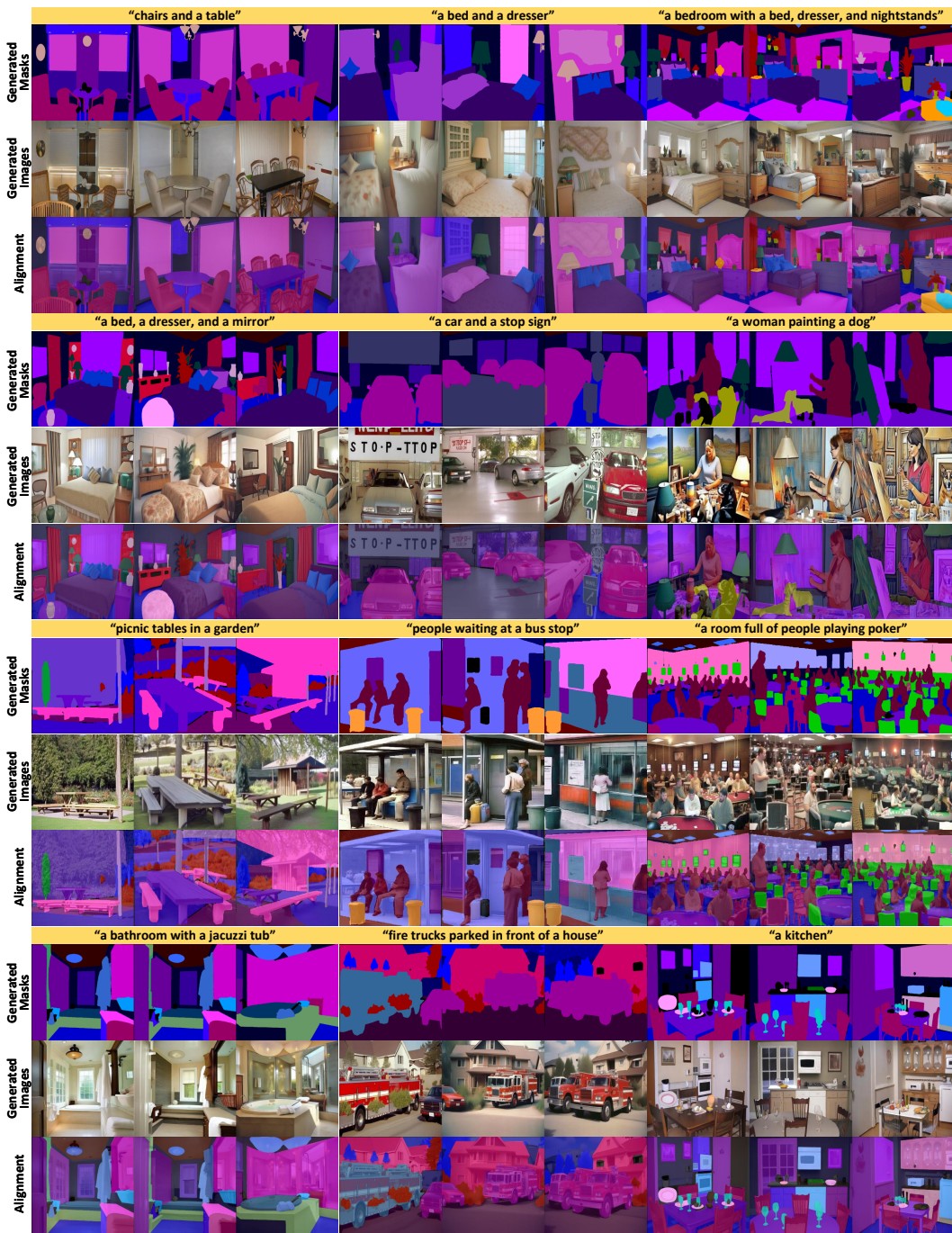

Figure 13: **More Generated Samples by MaskSyn on ADE20K:** Our synthetic segmentation masks are highly diverse, and the synthetic images align well with the respective masks and text prompts.

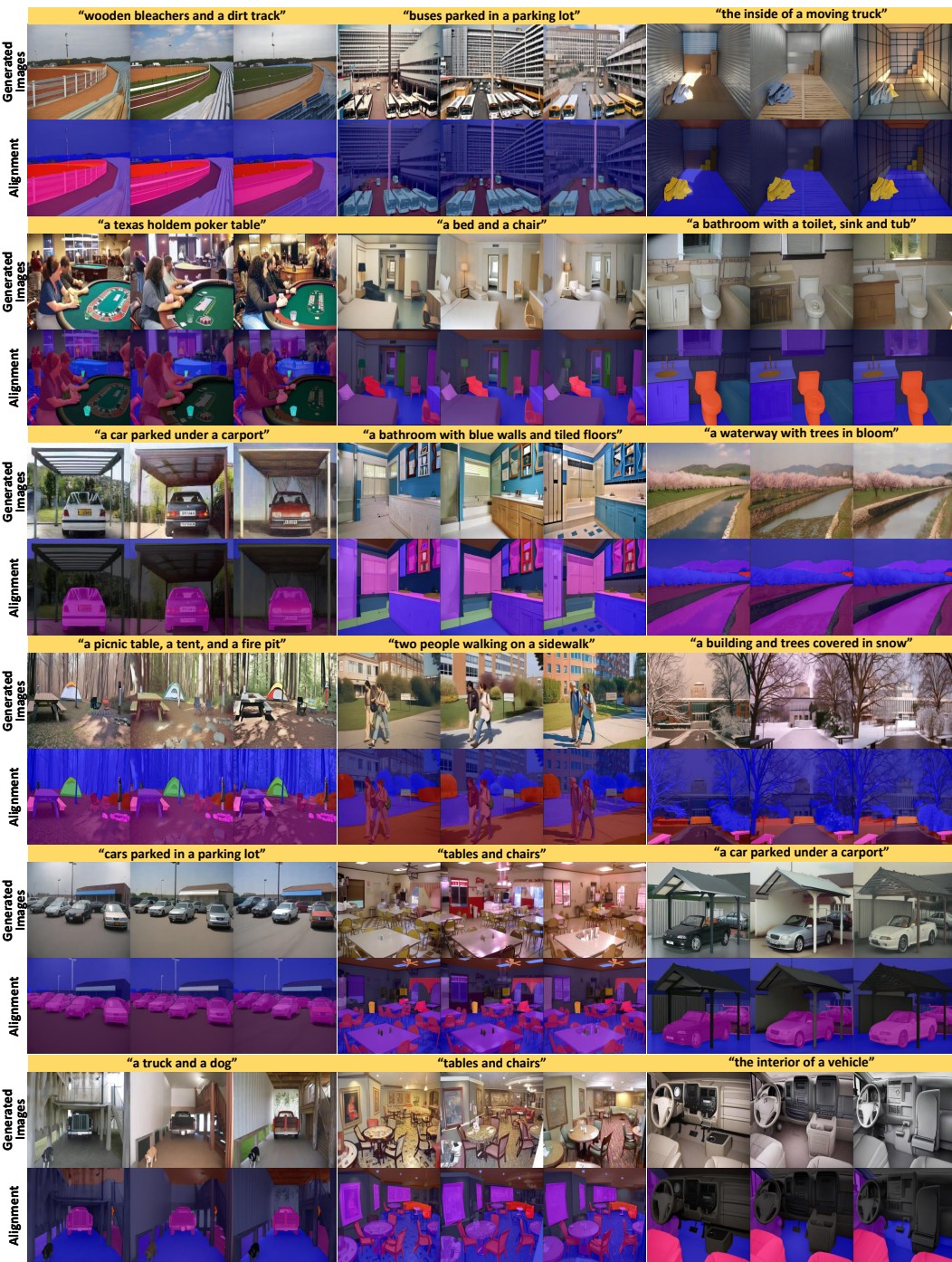

Figure 14: **More Generated Samples by ImgSyn on ADE20K:** The generated images align well with the text prompts and human-annotated segmentation masks.

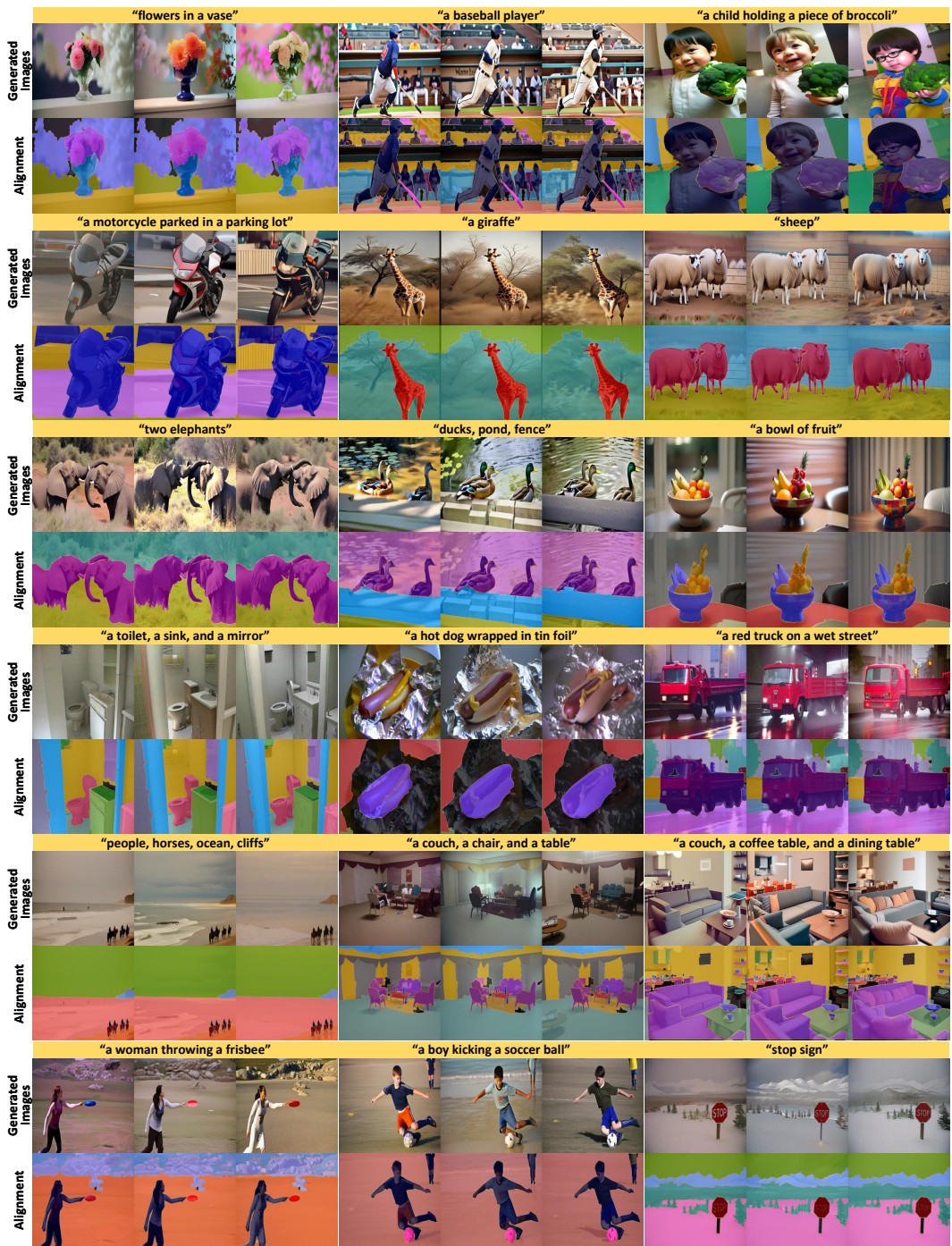

Figure 15: **More Generated Samples by ImgSyn on COCO:** The generated images align well with the text prompts and human-annotated segmentation masks.

