# OpenReview forum: "SegGen: Supercharging Segmentation Models with Text2Mask and Mask2Img Synthesis"
_ICLR.cc/2024/Conference — Submitted to ICLR 2024_

### Official Review · Reviewer_wtDG · 2023-10-28

**Soundness:** 3 good
**Presentation:** 3 good
**Contribution:** 3 good
**Rating:** 6
**Confidence:** 4

**Summary:**

This proposes SegGen, a highly-effective training data generation method for image segmentation, including two data generation strategies: MaskSyn and ImgSyn. Experiments show the synthetic data can be used for improving segmentation performance.

**Strengths:**

- The proposed method is resonable and easy to understand.
- The paper writing is good.
- The proposed method has improvements on multiple datasets.

**Weaknesses:**

- Two different training strategies, including synthetic pre-training and synthetic augmentation, seems not giving consistent improvement on different tasks. Synthetic augmentation seems giving performance drop on COCO dataset (i.e., 51.3 vs 52.0). It is not clear why sythetic augmentation is not good for training.
- It is better to compare synethic augmentation with the original data augmentation strategies.
- It is better to give the details of real data and synthetic data used in the proposed methods, which can help readers to understand the details.
- It is better to give the impact of different number of synthetic data in two different modules MaskSyn and ImgSyn.

**Questions:**

see weakness

---

> ### Author Response · Authors · 2023-11-21
> **Response to Reviewer wtDG**
>
> **Q1: Why is the synthetic pre-training strategy better than the synthetic augmentation strategy on COCO?**
>
> (1) We have provided related analysis on using different synthetic data training strategies on ADE20K and COCO in the ablation study of training strategies in Sec. 4.4. We believe that the two primary factors influencing the results are overfitting and domain gap.
>
> (2) On COCO, which has a training set of around 100,000 images with coarser annotations, overfitting is less of a concern. Instead, the domain gap between synthetic data and COCO data is a more significant factor. Our analysis in Figure 5 highlights the well-known annotation bias issue [1] in the COCO segmentation dataset, where many of our generated samples exhibit noticeably better alignment than human annotations. This is aligned with previous work [1] (Table 1 of [1]) which also found that the mask annotation quality of COCO is significantly worse than ADE20K. Consequently, in the case of COCO, our synthetic data proves more effective as a pre-training dataset, aiding the model in learning good initialization weights, which subsequently leads to improved performance metrics after fine-tuning on the human-annotated dataset.
>
> (3) With ADE20K, due to the limited size of the training split, which contains merely ∼20k images, segmentation models easily overfit the scarcely-available training images and segmentation masks. Utilizing our high-quality synthetic data for data augmentation can help largely mitigate such an overfitting problem therefore leading to significant performance improvement, as supported by Figure 11. Due to the high-quality annotation on ADE20K, the domain gap between our synthetic data and real data is relatively smaller.
>
>
> **Q2: Comparison with the original data augmentation strategies.**
>
> Our proposal is a data generation method, which has no conflict with the data augmentation methods including color augmentation, random flipping, and random cropping. To further demonstrate the effectiveness of our method, as suggested by the reviewer, we conduct an experiment to remove the color augmentation, random flipping, and random cropping of Mask2Former R50, and train it with our SegGen data on ADE20K. The results are shown in the table below. We find that our SegGen can significantly boost the mIoU by +4.8 when common data augmentation techniques are not employed. These remarkable results further confirm the effectiveness of our method in mitigating overfitting issues on the ADE20K dataset.
>
>
> |  Mask2Former R50 | Iterations  |  mIoU |
> |---|---|---|
> | Baseline w/o Data Aug | 160K  | 41.4   |
> | --w/ SegGen  |  160K | **46.2 (+4.8)**  |
> | Baseline w/ Data Aug | 160K  |  47.2  |
> | --w/ SegGen  |  160K |  **49.9 (+2.7)** |
>
>
>
> **Q3: Details of real data and synthetic data used in the proposed methods.**
>
> Thank you for your feedback. We have introduced the implementation details of our SegGen in Sec. 4.1. For real data, we utilize the whole training split of the ADE20K semantic segmentation dataset and the COCO panoptic segmentation/instance segmentation dataset to train our generative models separately. Taking ADE20K as an example, we use its training split (20,210 image-mask pairs) to train the text-to-mask and mask-to-image generation models. After training, for each training sample in the real dataset, we produce 10 synthetic mask-image pairs using MaskSyn, resulting in 202,100 training samples. Additionally, we synthesize 50 images based on each human-labeled mask with ImgSyn, leading to a total of 1,010,500 samples. In the training of segmentation models, we use all human-annotated data of ADE20K as well as our synthetic data, following the synthetic data augmentation strategy that we propose.
>
> **Q4: The impact of different numbers of synthetic data in two different modules MaskSyn and ImgSyn.**
>
> In Section 4.4 and Figure 8, we provide the experimental results of utilizing various numbers of ImgSyn samples. It is observed that the performance becomes stronger as we increase the amount of synthetic data. In accordance with the reviewer's suggestion, we conduct a series of experiments to examine the influence of using different numbers of MaskSyn samples, as shown in the table below. Similar to the observations in Figure 8 for ImgSyn, we achieve significantly improved performance after augmenting the quantity of MaskSyn samples. This further underscores the importance of generating large-scale segmentation data to train more effective segmentation models.
>
> |  Method | Iterations  |  mIoU |
> |---|---|---|
> | Baseline | 160K  |47.2 |
> | MaskSyn 40K | 160K  |  47.3   |
> | MaskSyn 80K  |  160K |  47.5 |
> | MaskSyn 120K | 160K  |47.6 |
> | MaskSyn 160K | 160K  |48.1 |
> | MaskSyn 200K | 160K  | **48.5** |
>
> [1] Agrim Gupta et al. LVIS: A Dataset for Large Vocabulary Instance Segmentation. CVPR 2019.

---

### Official Review · Reviewer_RiQa · 2023-11-01

**Soundness:** 4 excellent
**Presentation:** 3 good
**Contribution:** 3 good
**Rating:** 8
**Confidence:** 5

**Summary:**

This paper presents an interesting approach to generating segmentation datasets via Text2Mask and Mask2Image.
The main contribution is the introduction and the design of Text2Mask. It's novel and effective in improving the conventional segmentation model's performance. I think this data-centric viewpoint is valuable in this era of large models.

**Strengths:**

+ Text2Mask is effective. It gives us a critical insight into the diversity of mask annotation. It brings good knowledge improvement.
+ Color2SegmentationMask is simple yet effective. Before I read it, I was thinking about how to convert a continuous color map to a discrete segmentation mask. Nearest matching is a good way to solve it.

**Weaknesses:**

- SegGen is trained on paired segmentation datasets based diffusion model. This means diffusion prior is used in the model w/ SegGen. For a fair comparison, other models considered diffusion prior should be compared. For example, the UNet of the diffusion model, as a segmentation model with the same training setting, can be used for comparison. It's better to see this result.
- The related works of data generation in other domains are absent. For example, Scalable Multi-Temporal Remote Sensing Change Data Generation via Simulating Stochastic Change Process (ICCV'23) is highly related to the topic of data generation. Authors can discuss it in the related work for more broader impacts.

Minors:
A2. the second should be  Projection from Color Maps to Segmentation Masks.

**Questions:**

N/A

---

> ### Author Response · Authors · 2023-11-21
> **Response to Reviewer RiQa**
>
> **Q1: Comparison with segmentation models using UNet of diffusion models.**
>
> Thank you for your suggestion. We did not compare our method with this kind of approach because it has not achieved state-of-the-art performance on the most widely recognized segmentation benchmarks and has a higher computational cost. Instead, our method has the capability to enhance the performance of the top-performing segmentation models and thus has an apparent performance superiority.
>
> In response to the reviewer's suggestion, we carry out additional experiments to compare our approach with segmentation models that utilize UNet. VPD [1] is one such model that employs UNet of Stable Diffusion 1.5 (SD-1.5) model as its backbone. We use the same SD-1.5  as the base model of our generative models, and synthesize 200K MaskSyn samples and 200K ImgSyn samples on ADE20K. We train Mask2Former Swin-L with our SegGen data using exactly the same training settings of VPD (80K iterations, crop size 512, batch size 16) and show the results in the table below. Our method shows significantly better performance and computational efficiency compared with VPD, which indicates that our data generation method may be a more promising way to design generative models for segmentation tasks.
>
> |  Method| Base Diffusion Model | Iterations  |  mIoU | FLOPs |
> |---|---|---|---|---|
> | VPD| SD-1.5 | 80K | 53.7   | 891G |
> | Mask2Former  /w **SegGen (ours)**  | SD-1.5 | 80K| 54.4 |  **403G** |
> | Mask2Former  /w **SegGen (ours)**  | SDXL | 80K|  **54.8** |  **403G** |
>
> **Q2: Related work and minor issues.**
>
> Thank you for your advice. We will discuss the suggested paper on remote sensing data generation and fix the typo in the appendix.
>
> [1] Wenliang Zhao et al. Unleashing Text-to-Image Diffusion Models for Visual Perception. ICCV 2023

---

### Official Review · Reviewer_vgUs · 2023-11-01

**Soundness:** 1 poor
**Presentation:** 2 fair
**Contribution:** 1 poor
**Rating:** 5
**Confidence:** 3

**Summary:**

This paper presents a data generation method with a diffusion-based model for sementic segmentation. The authors created SegGen, a method that makes image segmenting better. It uses two ways to create training data: MaskSyn and ImgSyn. MaskSyn makes new image pairs to help train the model, while ImgSyn makes new images from existing ones. This makes their models perform even better on popular tests like ADE20K and COCO. For example, they achieved improvements in MIoU of ADE20K from 47.2 to 49.9.

**Strengths:**

This papers confirmed that SegGen could boost the performance of semantic segmentation by augmenting training datasets with images and masks generated by the fine-tuned SDXL regarding both ADE20K, COCO panatopic and COCO instance segmentation.

The paper is easy to follow and detailed. The expriments are quite comprehensive. Many example images help readers understand.

**Weaknesses:**

Unfortunately the novelty of the proposed idea is very low. Some works on generating training images for segmantic segmentation with generative models is proposed before. The idea of SegGen is not novel. Regarding MaskSyn and ImgSyn, SDXL-based was fine-tuned with ControlNet. All the components adopted in this paper are existing ones. In this sense, although the reviewer agrees with the effectiveness of the SegGen method, its novelty is not enough for the ICLR paper. The reviewer guesses that large parts of the performancs improvement comes from high-quality diffusion-based image generation model, SDXL. Since SDXL can generate realistic images which are hard to discriminate as fake images, the peformance must be much improved.

**Questions:**

What if the older SD model like V1 or other less-quality image generation models was used ? The reviewer would like to know the extent to which developments in image generation models have contributed to the success of this method.

---

> ### Author Response · Authors · 2023-11-21
> **Response to Reviewer vgUs**
>
> **Q1: There are some works on generation for segmentation. What is our difference and novelty?**
>
> Our method makes significant contributions in three key aspects: innovative methodology, unique generation model, and significant performance. We will discuss these three points in the following:
>
> (1) **Innovative methodology.** Our SegGen is a ground-breaking methodology that **solves a significant "chicken or egg dilemma" challenge in previous works** [1,2,3]: In these previous methods, they rely on small-scale segmentation/affinity networks as annotators to produce pseudo-masks for the synthesized data. However, when these low-quality synthetic masks are used to train segmentation models, the resulting performance cannot surpass the annotator networks. Consequently, they are unable to enhance the segmentation performance on the most challenging benchmarks. This can be seen as a   "chicken or egg dilemma". In our novel proposal, we first use text to generate diverse segmentation masks, and then use the masks to synthesize the corresponding realistic images. The effectiveness of our generation method is not bounded by the segmentation network performance. Thus, our method avoids the "chicken or egg dilemma" and is able to produce very high-quality synthetic data, as visualized in Figures 3, 4, 5, 13, 14, and 15.
>
> (2) **Unique generation model.** We make a significant contribution by introducing the **first-ever Text-to-Mask generation model**. Our Text2Mask model stands out from the rest, as it generates multiple masks simultaenously in a spatially dense manner, instead of generating just a single mask. For Text2Mask model, we devised an effective method to convert continuous class maps into discrete ones, which is something noted by reviewer RiQa as non-trivial (i.e., reviewer RiQa was figuring out how to do it prior to reading the paper).
>
> (3) **Significant performance.** The high-quality synthetic data generated by our method yields significant performance improvement compared with previous segmentation data generation methods [1,2,3].  Specifically, our method obtains a remarkable **+25.2 mIoU** improvement on ADE20K (Table 5 in the paper) compared with the previous best segmentation data generation method DiffuMask [1] (ICCV 2023) using the same segmentation model. DiffuMask cannot improve the state-of-the-art performance on the challenging ADE20K dataset, while ours obtains a significant gain of +2.7 mIoU. Furthermore, we are the **first data generation method that achieves significant performance improvements for top-performing segmentation models (Mask2Former, MaskDINO) on highly challenging semantic/panoptic/instance segmentation benchmarks including ADE20K and COCO.**
>
> **Q2: Does the performance gain result from the usage of SDXL? What is the performance if we use SDv1 as the base generative model?**
>
> (1) SDXL [4], a prevalent open-source image generation model, does not possess the capability to generate synthetic segmentation masks. Consequently, it is evident that SDXL alone would not lead to any performance improvements in segmentation tasks. It is our SegGen that is able to generate high-quality segmentation training data, which ultimately leads to significant enhancements in state-of-the-art segmentation performance.
>
> (2) Following the suggestion of the reviewer, we conduct an extended experiment on ADE20K to use SD-1.5 [5] as the base model in our text-to-mask and mask-to-image generation models. We synthesize 200K MaskSyn samples and 200K ImgSyn samples. We find that the generated samples maintain a similarly high quality compared to the ones generated by our previous models using SDXL as base model. We then use the newly generated data to train Mask2Former R50 on ADE20K alongside the real data, and show the results in the table below. Remarkably, our SegGen shows a significant performance gain of **+2.2 mIoU**. The performance improvement is close to our SDXL version (+2.7 mIoU), which demonstrates the strong robustness of our SegGen framework. On the other hand, the ability of SegGen to conveniently leverage advancements in image generation models is indeed a valuable advantage.
>
>
> |  Mask2Former R50| Iterations  |  mIoU |
> |---|---|---|
> | Baseline | 160K  |   47.2  |
> | --w/ SegGen MaskSyn  |  160K |   48.3 |
> | --w/ SegGen ImgSyn| 160K  |  48.9  |
> | --w/ SegGen MaskSyn + ImgSyn|  160K |  **49.4** |
>
>
> [1] Yuxuan Zhang et al. Datasetgan: Efficient labeled data factory with minimal human effort. CVPR 2021.
>
> [2] Jiahao Xie et al. Mosaicfusion: Diffusion models as data augmenters for large vocabulary instance segmentation. arXiv 2023.
>
> [3] Weijia Wu et al. Diffumask: Synthesizing images with pixel-level annotations for semantic segmentation using diffusion models. ICCV2023.
>
> [4] Dustin Podell et al. Sdxl: Improving latent diffusion models for high-resolution image synthesis. arXiv 2023.
>
> [5] Robin Rombach et al. High-resolution image synthesis with latent diffusion models. CVPR 2022.

---

### Official Review · Reviewer_vvk8 · 2023-11-03

**Soundness:** 3 good
**Presentation:** 4 excellent
**Contribution:** 3 good
**Rating:** 5
**Confidence:** 4

**Summary:**

This work studies the usage of image synthesis techniques to augment the training data for image segmentation models. More specifically, the authors proposed two techniques, one is text2mask synthesis and the other one is mask2image synthesis. The authors employed state-of-the-art open-sourced stable diffusion model for text-to-mask generation, and then ControlNet pipeline for mask-to-image generation. Afterward, two models are finetuned on COCO annotations and then used for augmenting the original COCO segmentation dataset. The experimental results demonstrate that the generated image-mask pairs are beneficial for COCO segmentation tasks under different settings. Further ablation studies shed light on how to use generated data to improve the image segmentation performance.

**Strengths:**

1. The authors explored a way of using generative models for data augmentation for training image segmentation models, With the increasing fidelity of image generation models, e.g., stable diffusion, it is worth to have a study on how the powerful image generation model can empower or benefit image understanding tasks.

2. This work proposed two strategies for augmenting image segmentation data to boost the performance of image segmentation models. Specifically, the authors employed stable diffusion model to generate masks from texts, and then used ControlNet-like model for mask-to-image generation. These two strategies together helps to attain a large amount of image-mask pairs with high quality.

3. Based on the proposed techniques, the authors augmented COCO training set for image segmentation and then tried two training strategies: pretraining and joint training. It turned out that the extra generated images are beneficial to the image segmentation models under different settings.

**Weaknesses:**

1. My main concern is about the relatively marginal improvement after adding a large amount of generated images. Though the author claimed over 2pt and 1pt gain for tiny and large models on ADE20k, I think the performance-price ratio is pretty low in that they generated over 1M images for ADE20K, which is significantly larger than the original training data size. Likewise, the authors also generated over 1M images for COCO, which is around 10 times bigger than the original training data. However, the gains for large models are less than 1pt across the board. These results make me highly doubt whether the generated images are really extrapolating the training data or just lazily interpolating the training data.

2. The ablation study using 1000 training examples is misleading. If I understand correctly, the authors still used the full COCO training set for training the image generation models. As such, the generation models are indeed able to "extrapolate" the training data beyond the 1000 examples, and the final improvements over the baseline which merely uses 1000 real examples are not surprising at all. I would suggest the authors fine-tune a generation model on the 1000 real examples and see whether the generation model can do "extrapolation", which I think is a very important factor in making the method shine.

3. There is no clear guidance on when we should use data augmentation training and when using the pretraining strategy with the generated data. The current study is somehow empirical and the audience cannot get a good amount of insights on when to use which techniques. In practice, however, having a principle is very important given that trial-and-error on the huge amount of generated data is usually with high cost. Moreover, Fig 7 implies another uncertainty on how to determine the augmentation probability on a custom dataset. The authors conducted the ablation study only on ADE20K, which is insufficient to draw any conclusion.

4. With all the above being said, I think the authors failed to study a very critical problem -- how to improve the diversity and quality of the generated images so that we can use much less generated samples to improve the performance of image segmentation models? Given what is presented in this paper, it is really hard to capture any idea about what we should do to improve the data-efficiency of the method, and what knowledge the generated images can bring to the image segmentation models.I would highly encourage the authors think about this and make the work more solid.

**Questions:**

Please see above comments.

---

> ### Author Response · Authors · 2023-11-21
> **Response to Reviewer vvk8 - Q1.1**
>
> **Q1.1: Is performance improvement significant?**
>
> (1) The significant improvement in performance achieved by our method is confirmed by unanimous agreement among all other reviewers (vgUs: “agrees with the effectiveness of the SegGen method”, RiQa: “effective”, wtDG: “highly-effective”).
>
> (2) Our method obtains a remarkable **+25.2 mIoU** (3 classes) improvement on ADE20K (Table 5 in the paper) compared with the previous best segmentation data generation method DiffuMask [1] (ICCV 2023) using the same segmentation model and purely synthetic data. DiffuMask cannot improve the state-of-the-art performance on the challenging ADE20K dataset, while ours obtains a significant gain of **+2.7 mIoU**.
>
> (3) It is important to note that ADE20 and COCO are the most common benchmarks for image segmentation research.  Many outstanding segmentation methods have been proposed to push metrics on these benchmarks to an exceptionally high level. Even so, our method still delivers substantial performance gains to the state-of-the-art Mask2Former model [2]:
> - **+2.7/+1.3 (R50/Swin-L) on ADE20K semantic segmentation,**
> - **+0.7/+0.7 (R50/Swin-L) on COCO panoptic segmentation,**
>
> while maintaining identical model structure and training computational costs. These compelling results firmly support the strength and effectiveness of our innovative approach.
>
> (4) Our performance gains are more significant than many recently published works on image segmentation. Specifically, when using Mask2Former (Swin-L) as the baseline,
> - Our method improves mIoU by **1.3** on ADE20K and PQ by **0.7** on COCO.
> - Mask DINO [3] (CVPR 2023) improves mIoU by 0.5  on ADE20K and PQ by 0.5  on COCO.
> - MP-Former [4] (CVPR 2023) improves mIoU by 0.8  on ADE20K, PQ by 0.3  on COCO.
> - DejaVu [5] (CVPR 2023) improves mIoU by 0.5  on ADE20K, and PQ by 0.4  on COCO.
>
>
> (5) What is more important, our data generation method can also be readily employed to boost the performance of more powerful segmentation models. Mask DINO [2] (CVPR 2023) is currently the best-performing segmentation model for COCO panoptic. Table 2 shows that applying our synthetic data to MaskDINO remarkably improves its performance on  COCO panoptic segmentation by **+0.7 PQ**. This compelling evidence further suggests the substantial performance gains achieved through our data generation method, leaving no doubt regarding its overwhelming impact.
>
>
> [1] Weijia Wu et al. Diffumask: Synthesizing images with pixel-level annotations for semantic segmentation using diffusion models. ICCV 2023.
>
> [2] Bowen Cheng et al. Masked-attention mask transformer for universal image segmentation. CVPR, 2022.
>
> [3] Feng Li et al. Mask dino: Towards a unified transformer-based framework for object detection and segmentation. CVPR 2023.
>
> [4] Hao Zhang et al. MP-Former: Mask-piloted transformer for image segmentation. CVPR 2023.
>
> [5] Shubhankar Borse et al. Dejavu: Conditional regenerative learning to enhance dense prediction. CVPR 2023.

---

> ### Author Response · Authors · 2023-11-21
> **Response to Reviewer vvk8 - Q1.2**
>
> **Q1.2: Can our generation models extend beyond mere interpolation of training data and effectively extrapolate information?**
>
> Yes, our method has strong extrapolation and generation ability for the following reasons:
>
> (1) **The strongest evidence is the significant performance gains** (as highlighted above in Q1.1) on the state-of-the-art segmentation models on well-recognized benchmarks. The reason is that if the model were merely synthesizing data that closely resembles the original samples, the segmentation models would suffer from overfitting and degraded performance since recent segmentation models are trained with extremely long training schedules.
>
> (2) Our method helps segmentation models obtain **superior generalization ability** for images from unseen domains (e.g. PASCAL and AIGC samples), as illustrated in Figure 1, 6, and 10. This clearly requires data extrapolation and cannot be achieved using mere data interpolation.
>
> (3) What gives our SegGen superior generation and extrapolation capabilities? We propose a novel data generation framework that **solves a very important "chicken or egg dilemma" issue in previous works** [1,2,3]. In the previous methods, they rely on small-scale segmentation/affinity networks as annotators to produce pseudo-masks for the synthesized data. However, when these low-quality synthetic masks are used to train segmentation models, the resulting performance cannot surpass the annotator networks. Consequently, they are unable to enhance the segmentation performance on the most challenging benchmarks. This issue is a  "chicken or egg dilemma". In our proposal, we first use text to generate diverse segmentation masks, and then use the masks to synthesize the corresponding realistic images. This ground-breaking generation framework cleverly avoids the "chicken or egg dilemma" and is able to produce very high-quality synthetic data, which transfers to significant performance improvement.
>
> [1] Weijia Wu et al. Diffumask: Synthesizing images with pixel-level annotations for semantic segmentation using diffusion models. ICCV 2023.
>
> [2] Yuxuan Zhang et al. Datasetgan: Efficient labeled data factory with minimal human effort. CVPR 2021.
>
> [3] Jiahao Xie et al. Mosaicfusion: Diffusion models as data augmenters for large vocabulary instance segmentation. arXiv 2023.

---

> ### Author Response · Authors · 2023-11-21
> **Response to Reviewer vvk8 - Q1.3**
>
> **Q1.3: Why do we need such a big synthetic dataset? How about the performance-price ratio?**
>
> (1) Firstly, it is critical to point out that **our method does not introduce additional computational overhead during segmentation model training**.  In our experiments, we employ identical training recipes for both the baseline models and ours, including iteration steps, batch size, and other training hyper-parameters. Specifically, for each training iteration, when we construct the training batch, we randomly replace the real samples with synthetic ones. This keeps the batch size consistent with the baseline, and our synthetic dataset shares the same computational training cost as the baseline. Our method only needs additional disk storage to store the synthetic data when training segmentation models, which is a negligible cost compared to the compute/GPU costs required for training segmentation models.
>
> (2) We hope that the creation of a large-scale, high-quality synthetic segmentation dataset through our method can be a useful contribution to the segmentation research community. Our synthetic dataset only has to be generated once and it can be leveraged by many different models. As evidenced by the results presented in Table 2 and 4, the inclusion of our high-quality synthetic segmentation data significantly enhances the performance of models with various architecture types  (e.g. Mask2Former, Mask DINO, and HRNet), and we believe it can also benefit other segmentation models. On the other hand, synthesizing data is naturally much more cost-effective than hiring humans to annotate an equivalent amount of data. With the cost advantage, there are no strong reasons for not generating a substantial amount of synthetic data for a thorough evaluation of its ultimate effectiveness on prevalent benchmarks (i.e. ADE20K and COCO).
>
> (3) We provide a study on the performance-price ratio in Figure 8. It shows the relationship between the size of the synthetic dataset and the resulting performance. Our method demonstrates a notable performance gain (+0.7 mIoU) even when there are only 20k synthetic images generated for the ADE20K dataset. We would like to point out that while the previous SOTA method DiffuMask [1] generated around 100K synthetic samples, the segmentation models trained with their synthetic data cannot even match the baseline performance on ADE20K. This finding validates the significant quality advantage of our data generation method. The satisfactory performance gain from our small synthetic dataset motivates us to scale up our synthetic data generation method to generate 1M image-mask pairs. Our final 1M synthetic dataset brings a much more significant performance gain of 2.7 mIoU on ADE20K. We consider this ablation experiment to be another important contribution of our paper, which informs the critical importance of the scale of synthetic data, akin to real data. In Appendix A.6 and Figure 11, we delve deeper into the analysis of why scale plays a vital role. Overfitting emerges as a major factor. When using only 20K synthetic samples, the segmentation model encounters each synthetic sample approximately 76.8 times during training. In contrast, with 1M synthetic samples, each sample is presented to the model approximately 1.5 times. Consequently, scaling up the synthetic dataset size significantly alleviates the risk of overfitting the models to the data.
>
> [1] Weijia Wu et al. Diffumask: Synthesizing images with pixel-level annotations for semantic segmentation using diffusion models. ICCV 2023.

---

> ### Author Response · Authors · 2023-11-21
> **Response to Reviewer vvk8 - Q2**
>
> **Q2: Ablation study using 1000 training examples.**
>
> (1) First of all, as mentioned above, the effectiveness and extrapolation ability of our method have been firmly confirmed by the significant performance improvement in the case of using all training data, which is a much more convincing proof given the extremely high difficulty of improving on the prevalent large-scale segmentation benchmarks. In Section 4.4, we explicitly state that "while we use only 1,000 real samples for synthetic data sampling, the generative models are trained on the complete training set."
>
> (2) Nonetheless, we agree with the reviewer that it is meaningful to examine the effectiveness of our method under the constraint of using only 1000 samples for training both the generative models and segmentation models. Therefore, we conduct an extended experiment wherein we utilize the same 1000 samples to train the text-to-mask and mask-to-image generation models. Subsequently, we generate 200K MaskSyn samples and 200K ImgSyn samples using these newly trained models. We employ these additional synthetic data to train Mask2Former R50 and HRNet W48 models with the 1000 real samples, and the outcomes are presented in the table provided below. In the experiments, we find that although the generation quality is constrained by the highly limited training data, both MaskSyn and ImgSyn samples still result in notable performance enhancements on the two different segmentation architectures. Using both MaskSyn and ImgSyn improves mIoU by +3.4 (Mask2Former) and +3.3 (HRNet), which demonstrates the undeniable extrapolation ability and data efficiency of our method.
>
> |  Mask2Former R50| Iterations  | Real Samples  |  mIoU |
> |---|---|---|---|
> | Baseline | 160K  | 1000  |  23.7 |
> | --w/ SegGen MaskSyn  |  160K |  1000   |  26.5 |
> | --w/ SegGen ImgSyn| 160K  |  1000   | 24.1  |
> | --w/ SegGen MaskSyn + ImgSyn|  160K | 1000    |  **27.1** |
> | Baseline | 160K  | 20210 |  47.2 |
> | --w/ SegGen MaskSyn  |  160K |  20210   |  48.5 |
> | --w/ SegGen ImgSyn| 160K  |  20210   | 49.3 |
> | --w/ SegGen MaskSyn + ImgSyn|  160K | 20210    |  **49.9** |
>
>
> |  HRNet W48| Iterations  | Real Samples  |  mIoU |
> |---|---|---|---|
> | Baseline | 160K  | 1000  |  23.6 |
> | --w/ SegGen MaskSyn  |  160K |  1000   | 26.0 |
> | --w/ SegGen ImgSyn| 160K  |  1000   | 23.9  |
> | --w/ SegGen MaskSyn + ImgSyn|  160K | 1000    | **26.9** |
> | Baseline | 160K  | 20210 |  44.4 |
> | --w/ SegGen MaskSyn  |  160K |  20210   |  44.8 |
> | --w/ SegGen ImgSyn| 160K  |  20210   | 45.4 |
> | --w/ SegGen MaskSyn + ImgSyn|  160K | 20210    |  **45.6** |

---

> ### Author Response · Authors · 2023-11-21
> **Response to Reviewer vvk8 - Q3**
>
> **Q3.1: Guidance on using synthetic augmentation or synthetic pretraining strategies.**
>
> (1) We have provided guidance and insight on using different synthetic data training strategies on ADE20K and COCO in the ablation study of training strategies in Sec. 4.4. We believe that the two primary factors influencing the results are overfitting and domain gap.
>
> (2) On ADE20K, due to the limited size of the training split, which contains merely ∼20k images, segmentation models easily overfit the scarcely-available training images and segmentation layouts. Utilizing our high-quality synthetic data for data augmentation can help largely mitigate such an overfitting problem therefore leading to significant performance improvement.
>
> (3) On the other hand, with COCO, which has a larger training set of around 100,000 images, overfitting is less of a concern. On COCO, we notice a significant domain gap between the synthetic data and the original COCO data. Our analysis in Figure 5 shows the annotation bias issue in the COCO segmentation dataset where many of our generated samples exhibit noticeably better mask alignment than human annotations on COCO. Table 1 of previous work [1] also found that the mask annotation quality of COCO is significantly worse than ADE20K. This annotation bias leads to a significant domain gap between COCO samples and our generated samples. Due to the large gap, it is unfavorable to jointly use our synthetic data and COCO samples at the same time for training segmentation models.  Therefore, in the case of COCO, our synthetic data proves more effective as a pre-training dataset, aiding the model in learning good pretrained weights. This helps the model obtain significantly stronger performance after fine-tuning on the COCO training set.
>
>
> **Q3.2 Is the trial cost using synthetic data higher than real data? How to determine the augmentation probability on a custom dataset?**
>
> (1) As previously highlighted in Q1.3, the use of our synthetic data does not result in an increase in computational cost for training segmentation models in terms of GPU memory, FLOPs, parameters, and other related factors. The only additional requirement is more disk storage, which is insignificant compared to the substantial compute/GPU cost required for training state-of-the-art segmentation models. Consequently, the cost of conducting trials using synthetic data is almost equivalent to the cost of regular training on real data.
>
> (2) For the augmentation probability, we would like to remind the reviewer that there are almost no data augmentation methods, such as random cropping, color jittering, mixup, etc., that can provide a theoretical guarantee on the optimal augmentation rates. The empirical nature of training modern deep learning models is widely recognized within the community. However, we have carefully examined its influence on ADE20K in Fig. 7. It is noteworthy that across a broad range of augmentation rates (**from 20% to 80%**), our synthetic data consistently enhances the performance of segmentation models compared with the baseline. This observation highlights the robustness of our segmentation generation method.
>
> [1] Agrim Gupta et al. LVIS: A Dataset for Large Vocabulary Instance Segmentation. CVPR 2019.

---

> ### Author Response · Authors · 2023-11-21
> **Response to Reviewer vvk8 - Q4**
>
> **Q4.1 Does our method improve the diversity and quality of the generated data?**
>
> (1) Actually, the most substantial contribution of this work lies in the exceptionally strong diversity and quality of the synthetic data generated through our method. This remarkable diversity and quality greatly surpass those of all previous methods [1-3], which is the key factor enabling us to be the **first data generation method that achieves significant performance improvements for top-performing segmentation models on highly challenging semantic/panoptic/instance segmentation benchmarks including ADE20K and COCO.**
>
> (2) To provide a more intuitive showcase of the diversity and quality of our generated data, we have presented visual samples in Figures 3, 4, 5, 13, 14, and 15. Especially, as illustrated by Fig. 5, our synthetic images even demonstrate higher quality than human-annotated data in many cases. This phenomenon, which has not been observed in previous works, serves as a significant milestone in the field of segmentation generation. It further highlights the groundbreaking nature of our approach. As a result, our approach surpasses the previous best-performing data generation method [3] on ADE20K with a remarkable +25.2 mIoU improvement when using only synthetic data, as shown in Table 5. In this sense, SegGen has the highest data efficiency compared with these methods.
>
> (3) Finally, we would like to emphasize that the extensive scale of synthetic data is a strength rather than a weakness. At large data scale, only high-quality synthetic data can enhance the performance of segmentation models, while low-quality data will only have a negative impact on performance.
>
> **Q4.2 What is the data efficiency of the proposed SegGen?**
>
> Being the first to be able to achieve substantial performance enhancements on highly challenging segmentation benchmarks, our method naturally exhibits significantly higher data efficiency compared to previous approaches in the field of generation for segmentation. For example, we achieve a remarkable +25.2 mIoU improvement compared with the previous best [3] on ADE20K when using only synthetic data.  On the other hand, as highlighted in our response to Q2, our method produces significant enhancements in segmentation models even when they are trained with only 1000 training samples. This further emphasizes the high data efficiency of our method in this regard.
>
>
> [1] Yuxuan Zhang et al. Datasetgan: Efficient labeled data factory with minimal human effort. CVPR 2021.
>
> [2] Jiahao Xie et al. Mosaicfusion: Diffusion models as data augmenters for large vocabulary instance segmentation. arXiv 2023.
>
> [3] Weijia Wu et al. Diffumask: Synthesizing images with pixel-level annotations for semantic segmentation using diffusion models. ICCV2023.

---

### Author Response · Authors · 2023-11-22
**General Responses to All Reviewers**

We would like to thank all the reviewers for their time and efforts to provide valuable feedback on our work. We have provided thorough explanations and supporting materials in response to each reviewer's concerns. It is our aim that these discussions adequately address all of your questions. Moreover, we appreciate that the reviewers find our segmentation data generation method "novel" (RiQa), "worthy" (vvk8), and "highly-effective" (wtDG), besides pointing out that our paper provides “comprehensive experiments” (vgUs). We are thrilled to share our findings with the research community. We encourage open discussion and welcome any additional questions you may have.

---

### Meta-Review · Area_Chair_Am7j · 2023-12-05

**Metareview:**

This paper proposed a simple method for using generative models to create segmentation supervision for training models.
The authors use StableDiffusion and ControlNet to achieve text to mask and mask to image generation.
The model is fine-tuned on the COCO annotations. The generated masks are used to supplement the training on SOTA segmentation models such as Mask2Former which improves their performance.
The authors also provided several additional experiments showing that this technique robustly improves other models such as MaskDINO.
Given that ControlNet conditions based on a mask resized to latent space of the model, I believe that the current approach will struggle with generating masks for small objects.
While the authors chose to study semantic segmentation, I wonder how this approach scales with data on open-vocabulary instance segmentation tasks. With the rise of large scale segmentation datasets, it is unclear how useful synthetic augmentation will be in that regime.
Finally, the gains that this method provides are small (also noted by Rvvk8) and also diminish for larger models -- 1% AP on ADE20K. I understand that the gains are larger for certain classes, but if that's the main contribution or result that the authors wish to highlight, it needs to be (a) studied/explained why these particular classes; (b) why this matters for other tasks beyond the 3 classes in ADE20K.

**Justification For Why Not Higher Score:**

Using generative models to augment data for training supervised models is not novel (done in many other domains).
It cleverly combines existing methods to achieve a good synthesis method.
However, the gains are small and diminish with larger models.

**Justification For Why Not Lower Score:**

N/A

---

### Decision · Program_Chairs · 2024-01-16

Reject